# COLLABORATIVE SYMMETRICITY EXPLOITATION FOR OFFLINE LEARNING OF HARDWARE DESIGN SOLVER

## ABSTRACT

This paper proposes *collaborative symmetricity exploitation* (CSE), a novel symmetric learning scheme of contextual policy for offline black-box placement problems. Leveraging the symmetricity increases data-efficiency by reducing the solution space, and improves generalization capability by capturing the invariant nature present regardless of changing context. To this end, we design a learning scheme that reduces the order bias (ex., neural network recognizes $\{1, 2, 3\}$ and $\{2, 1, 3\}$ as difference placement design) inherited from a sequential decision-making scheme of neural policy by imposing action-permutation (AP)-symmetricity (i.e, the permuted sequences are symmetric-placement of the original sequence) of placement problems. We first defined the order bias and proved that AP-symmetricity is imposed when the order bias of neural policy becomes zero. Then, we designed two collaborative losses for learning neural policy with reduced order bias: expert exploitation and self-exploitation. The expert exploitation loss is designed to clone the behavior of the expert solutions considering order bias. The self-exploitation loss is designed to be a special form of order bias where it measures AP-symmetricity from a self-generated solution. CSE is applied to the decoupling capacitor placement problem (DPP) benchmark, a significant offline black-box placement design problem in hardware domain that requires contextual policy. Experiments show that CSE outperforms state-of-the-art solver for the DPP benchmark.

## 1 INTRODUCTION

With the CMOS technology shrinking and increasing data rate, the design complexity of very large-scale integrated (VLSI) has increased. Human experts are no longer able to design hardware without the help of electrical design automation (EDA) tools, and EDA tools now suffer from long simulation time and insufficient computing power, making machine learning (ML) application to hardware design inevitable. Many studies have already shown that deep reinforcement learning (DRL), one of the representative ML methods for sequential decision making, is promising in various tasks in modern chip design; chip placement (Mirhoseini et al., 2021; Agnesina et al., 2020), routing (Liao et al., 2019; 2020), circuit design (Zhao & Zhang, 2020), logic synthesis (Hosny et al., 2020; Haaswijk et al., 2018) and bi-level hardware optimization (Cheng & Yan, 2021).

However, most previous DRL-based hardware design methods do not take the following into consideration. (a) Online simulators for hardware are usually time intensive and inaccurate; thus, learning with existing offline data by experts is more reliable. Since there exists a limited number of offline hardware data, a data-efficient learning scheme is necessary. (b) Hardware design is composed of electrically coupled multi-level tasks where task conditions are determined by the design of higher-level tasks; thus, a solver (i.e., contextualized policy conditioned by higher-level tasks) with high generalization capability to adapt to varying task conditions is necessary.

In this paper, we leverage the solution symmetricity of placement problem for data efficiency and generalization capability. Conventional sequential decision-making schemes for placement problems (Park et al., 2020; Mirhoseini et al., 2021; Cheng & Yan, 2021) auto-regressively generate solutions without considering the solution symmetricity, thus having the order bias; the neural network identifies the action-permutation (AP) symmetric solutions (i.e. identical placement designs), for instance, $\{1, 2, 3\}$ and $\{2, 1, 3\}$, as different solutions. Our proposed method overcomes the order bias limitation of the previous sequential decision-making schemes with a novel regularization

technique. Tackling the order bias (i.e. inducing AP-symmetricity) improves the data efficiency of training and generalization capability of the trained policy due to the two following reasons. First, data efficiency in training can be improved as learning the AP-symmetricity reduces the exploration space (see Fig. 1); neural network can automatically learn not only from the explored trajectories but also from their symmetric solution trajectories without additional exploration and simulation. Second, generalization capability on task variation can be improved as AP-symmetricity is the task-agnostic nature of placement problems.

To this end, we devised *collaborative symmetricity exploitation* (CSE) framework, a simple but effective method to induce AP-symmetricity with two collaborative learning schemes: expert exploitation and self-exploitation. The expert exploitation simply augments the offline expert data (sequential data) with a random permutation operator and uses it for imitation learning. The self-exploitation generates pseudo-labeled solutions from the current training policy, transforms the pseudo-labeled solution with a random permutation operator, and forces the solver to have an identical probability to generate the original pseudo-labeled solution and the transformed solution.

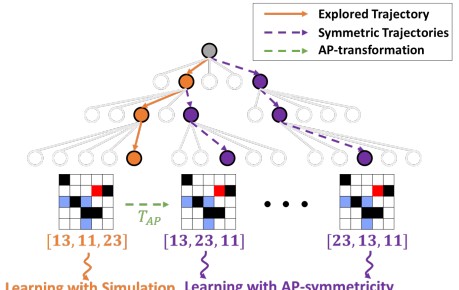

Figure 1: Conventional sequential decision-making method's heterogeneous trajectories from AP-Symmetric solution group.

To verify the effectiveness of CSE, we applied CSE to the decoupling capacitance (decap) placement problem (DPP), one of the significant hardware design benchmarks. The objective of DPP is to place a given number of decaps on power distribution network (PDN) with two varying conditions: keep-out regions and probing port location, determined by higher-level problems such as chip placement and routing. The goal of CSE is to train a solver (i.e., contextualized policy) that has high generalization capability to any given task condition.

**Contribution 1: A novel symmetric learning scheme for contextualized policy.** There exists several works (Cohen & Welling, 2016; Thomas et al., 2018; Fuchs et al., 2020; Satorras et al., 2021) that learn various symmetricities of input data in the domain space for regression and classification tasks. However, learning symmetricity in solution space is less studied as learning the symmertrcities in solution space of sequential policy (generative decision) is challenging. Bengio et al. (2021) tackled solution symmetricity of sequential policy by turning the Markov decision process (MDP) tree model into the directed acyclic graph (DAG)-based flow model. However, they target single-task optimization where the optimal solution set is unchanged. On the other hand, our CSE is an effective solution symmetric learning scheme for the contextualized policy capable of adapting to newly given task-condition.

**Contribution 2: DPP benchmark release.** DPP is a widely studied task in hardware domain without public release of the simulation models and source codes for the methods. Also, DPP can be seen as a contextual offline black-box optimization benchmark with extended properties compared to the design-bench (Trabucco et al., 2022), a representative non-contextual offline black-box optimization benchmark. In this work, by releasing the DPP benchmark with open-source simulation models and our reproduced baselines, DRL-based methods, meta-heuristic methods, behavior cloning-based methods, and our state-of-the-art CSE method, we expect huge industrial impacts on the hardware and the ML communities.

## 2 DECAP PLACEMENT PROBLEM (DPP) FORMULATION

This paper seeks to solve the decoupling capacitor placement problem (DPP), one of the essential hardware design problems. Decoupling capacitor (decap) is a hardware component that reduces power noise along the power distribution network (PDN) of hardware devices and improves the power integrity (PI). With transistor scaling and continuously decreasing supply voltage margin (Hwang et al., 2021), power noise has become a huge technical bottleneck in high-speed computing systems. Generally, the more decaps are placed, the more reliable the power supply is. However, adding more decaps requires more space and is costly. Thus, finding an optimal placement of decaps is essential in terms of hardware performance and cost/space-saving.

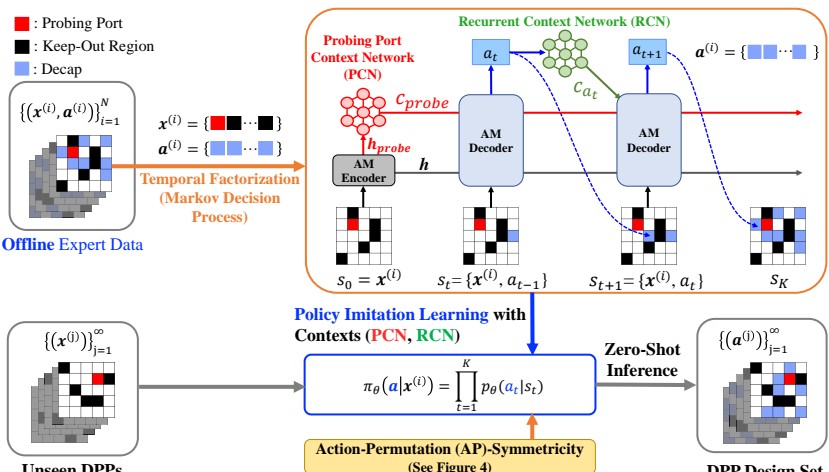

Figure 2: Overall pipeline of DPP contextualized policy parameterization with offline learning.

The goal of DPP benchmark is to optimally place a pre-defined number of decaps on a target PDN, given two conditions determined by higher-level tasks. First, keep-out regions are action-restricted areas where decaps cannot be placed as a design constraint. Second, probing port is the target chip/logic block location where the objective, power integrity (PI), is evaluated. The ports on PDN are represented as $N_{row} \times N_{col}$ grids, and the number of decaps is denoted by $K$. See Appendix A.2 for details of PDN and decap modeling for the benchmark.

Remark that DPP cannot be formulated as a conventional mixed-integer linear programming (MILP)-based combinatorial optimization because PI performance can not be formulated as a closed analytical form but can only be measured or simulated. This study aims to learn an effective DPP solver that can be used in practice.

## 2.1 CONTEXTUAL MARKOV DECISION PROCESS (MDP) OF DPP

As shown in Fig. 2, the procedure for solving DPP is modeled as a contextual Markov decision process (CMDP). CMDP is an augmented MDP proposed by (Hallak et al., 2015). The parameters of CMDP, transition and reward function, change based on the context variable. DPP can be formulated as CMDP, where the objective function $\mathcal{J}(; \boldsymbol{x})$ is determined by the task-condition $\boldsymbol{x} \in \mathcal{X}$ (i.e. context); the $\mathcal{X}$ is task-condition space.

Specifically, our objective function $\mathcal{J}(; \boldsymbol{x})$ is determined by PDN which is contextualized by $\boldsymbol{x}$. The contextualized PDN is represented as a set of three-dimensional feature vectors $\boldsymbol{x} = \{\boldsymbol{x}_i\}_{i=1}^{N_{row} \times N_{col}}$, where each grid (i.e., port) on PDN is represented as $\boldsymbol{x}_i = (x_i, y_i, c_i)$, in which $x_i, y_i$ indicate 2D coordinates of location, $c_i$ indicates the condition of port whether it belongs to a probing port $I_{probe}$ ($c_i = 2$), keep-out regions $I_{keepout}$ ($c_i = 1$), or decap allowed ports $I_{allowed}$ ($c_i = 0$). Note that $I_{keepout}$ and $I_{allowed}$ represent index sets consisting of indices corresponding to keep-out regions and decap allowed ports, respectively. $I_{probe}$ refers to an index of probing port. See Appendix A.3.

The design process sequentially places decaps on the available PDN ports until planning all the designated $K$ decaps. We model this CMDP with state, action, and policy as follows:

**State** $\boldsymbol{s}_t$ contains task-condition $\boldsymbol{x}$ and previous selected actions: $\boldsymbol{s}_t = \{\boldsymbol{x}, \boldsymbol{a}_{1:t-1}\}$.

**Action** $a_t \in \{1, ..., N_{row} \times N_{col}\} \setminus \boldsymbol{s}_{t-1}$ is defined as an allocation of a decap to one of the available ports on PDN. The available ports are the ports on PDN except for the probing port, keep-out ports, and the previously selected ports. The concatenation of sequentially selected actions $\boldsymbol{a} = a_{1:K}$ indicates the final decap placement *solution*.

**Policy** $\pi_\theta(\boldsymbol{a}|\boldsymbol{x})$ is the probability of producing a specific solution $\boldsymbol{a} = \boldsymbol{a}_{1:K}$, given task-condition $\boldsymbol{x}$, and is factorized as:

$$\pi_\theta(\boldsymbol{a}|\boldsymbol{x}) = \prod_{t=1}^{K} p_\theta(a_t|\boldsymbol{s_t}), \tag{1}$$

where $p_\theta(a_t|\boldsymbol{s_t})$ is the segmented one-step action policy parameterized by the neural network.

The objective of DPP is to find the optimal parameter $\theta^*$ of the policy $\pi_\theta(\cdot|\boldsymbol{x})$ as:

$$\theta^* = \arg\max_\theta \mathbb{E}_{\boldsymbol{x} \sim p_\mathcal{X}(\boldsymbol{x})} \mathbb{E}_{\boldsymbol{a} \sim \pi_\theta(\cdot|\boldsymbol{x})} [\mathcal{J}(\boldsymbol{a}; \boldsymbol{x})], \tag{2}$$

where $p_\mathcal{X}(\boldsymbol{x})$ is the probability distribution for varying *task-condition* $\boldsymbol{x}$ and $\mathcal{J}$ is objective function. Finding the optimal policy for various DPPs is a task contextual learning problem, in which each DPP has a distinct task condition. Once the task $\boldsymbol{x}$ is specified by $p_\mathcal{X}(\boldsymbol{x})$, the state-action space with complexity of $\binom{N_{row} \times N_{col} - 1 - |I_{keepout}|}{K}$ is determined. Thus, an efficient policy $\pi_\theta(\boldsymbol{a}|\boldsymbol{x})$ should capture the contextual features among varying task conditions $\boldsymbol{x}$.

Note that CSE is based on imitation learning, by which objective function $\mathcal{J}$ is implicitly induced through offline expert data. The objective function of DPP is described in equation 3.

## 2.2 Objective Function of DPP

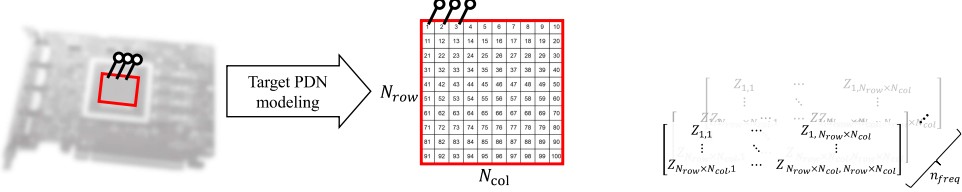

(a) Unit-cell representation of target PDN of a hardware device      (b) Z-parameter of target PDN.

Figure 3: Unit-cell and Z-parameter representations of real-world target PDN.

**Performance Evaluation Metric.** The performance of DPP is evaluated by power integrity (PI) simulation that computes the level of impedance suppression over a specified frequency domain and is quantified as:

$$\mathcal{J} := \sum_{f \in F} (Z_{initial}(f) - Z_{final}(f)) \cdot \frac{1\text{GHz}}{f} \tag{3}$$

where $Z_{initial}$ and $Z_{final}$ are the initial and final impedance at the frequency $f$ before and after placing decaps, respectively. $F$ is the set of specified frequency points. As shown in Fig. 3, the PI simulation for $(N_{row} \times N_{col})$ PDN requires a $N_{row}N_{col} \times N_{row}N_{col} \times n_{freq}$ (number of frequency points) sized Z-parameter matrix calculation because each port is electrically coupled to the rest of the ports and Z (i.e., impedance) is frequency-dependent. Thus, performance evaluation with a large Z-parameter matrix calculation is costly. The more impedance is suppressed, the better the power integrity and the higher the performance score. Note that this performance metric was also used for collecting the offline expert data using a genetic algorithm (GA).

## 3 Methodology

This section provides technical details of the proposed *collaborative symmetricity exploitation* (CSE) framework and the modified attention model (AM) neural architecture with two domain-specifically devised context neural networks for training a DPP solver.

### 3.1 Action-permutation Symmetricity and Order Bias of Placement Task

The symmetricity found in placement problems is the action-permutation (AP)-symmetricity, the order of placement does not affect the design performance. Let us denote $t_i$ a permutation of an action sequence $\{1, ..., K\}$, where $K$ is the length of the action sequence. We then define the AP-transformation $T_{AP} = \{t_i\}_{i=1}^{K!}$ as a set of all possible permutations. The AP-symmetricity of DPP is induced to the learned solver through the AP-transformation $T_{AP}$.

**Definition 3.1** (AP-symmetricity). For any $\boldsymbol{a} \in \mathcal{A}, \boldsymbol{x} \in \mathcal{X}, t \in T_{AP}$ where $\mathcal{A}$ is solution space and $\mathcal{X}$ is task-condition space,

- Scala-valued Function $f : \mathcal{A} \times \mathcal{X} \to \mathbb{R}$ is AP-symmetric if, $f(\boldsymbol{a}, \boldsymbol{x}) = f(t(\boldsymbol{a}), \boldsymbol{x})$.
- Conditional probability $\pi$ is AP-symmetric if, $\pi(\boldsymbol{a}|\boldsymbol{x}) = \pi(t(\boldsymbol{a})|\boldsymbol{x})$.

The objective function $\mathcal{J} : \mathcal{A} \times \mathcal{X} \rightarrow \mathbb{R}$ of DPP is an AP-symmetric function because $\boldsymbol{a}$ and $t(\boldsymbol{a})$ have identical placement design. The main role of CSE is to induce AP-symmetricity to the policy $\pi$ (conditional probability) to reflect the AP-symmetricity of an objective function $\mathcal{J}$.

Moreover, we define an order bias metric, $b(\pi; \boldsymbol{p})$, to measure AP-symmetricity.

**Definition 3.2 (Order bias on distributions $\boldsymbol{p} = \{p_{\mathcal{X}}, p_{\mathcal{A}}, p_{T_{AP}}\}$).** For a conditional probability $\pi(\boldsymbol{a}|\boldsymbol{x})$, where $\boldsymbol{x} \in \mathcal{X}$ (task-condition space) and $\boldsymbol{a} \in \mathcal{A}$ (solution space), the order bias $b(\pi; \boldsymbol{p})$ is defined as:

$$b(\pi; \boldsymbol{p}) = \mathbb{E}_{\boldsymbol{x} \sim p_{\mathcal{X}}(x)} \mathbb{E}_{\boldsymbol{a} \sim p_{\mathcal{A}}(\boldsymbol{a})} \mathbb{E}_{t \sim p_{T_{AP}}(t)} [||\pi(\boldsymbol{a}|\boldsymbol{x}) - \pi(t(\boldsymbol{a})|\boldsymbol{x})||_1]$$

Intuitively, the order bias $b(\pi; \boldsymbol{p})$ is a general property of a sequential solution generation scheme. It measures how much the solver $\pi(\boldsymbol{a}|\boldsymbol{x})$ has different probabilities to generate AP-symmetric solutions. The order bias metric holds for the following theorem:

**Theorem 3.1.** Task-conditioned policy $\pi(\boldsymbol{a}|\boldsymbol{x})$ is AP-symmetric if and only if order bias is zero $(b(\pi; \boldsymbol{p}) = 0)$ while the distributions are non-zero, $p_{\mathcal{X}}(\boldsymbol{x}) > 0, p_{\mathcal{A}}(\boldsymbol{a}) > 0, p_{T_{AP}}(t) > 0$, for any $x \in \mathcal{X}, \boldsymbol{a} \in \mathcal{A}$ and $t \in T_{AP}$. **Proof.** *See Appendix G.*

### 3.2 COLLABORATIVE SYMMETRICITY EXPLOITATION (CSE) FRAMEWORK

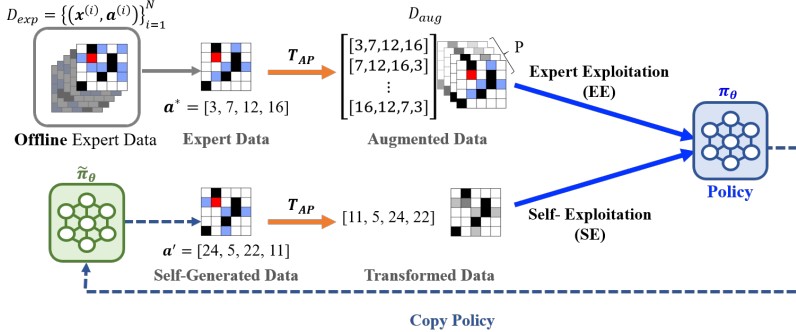

Figure 4: Illustration of collaborative symmetricity exploitation (CSE) process.

The CSE framework was designed to induce the AP-symmetricity to the trained model to improve the generalization capability and to allow data-efficiency in training.

To train a contextualized policy with a limited number of expert data, we designed the CSE loss term $\mathcal{L}$ consisting of expert exploitation loss $\mathcal{L}_{Expert}$ and self-exploitation loss $\mathcal{L}_{Self}$. Each loss function is mainly designed to reduce bias order (Definition 3.2):

$$\mathcal{L} := \mathcal{L}_{Expert} + \lambda \mathcal{L}_{Self} \tag{4}$$

$$\mathcal{L}_{Expert} = -\mathbb{E}_{\boldsymbol{a}^*, \boldsymbol{x} \sim D_{aug}}[log \pi_\theta(\boldsymbol{a}^*|\boldsymbol{x})] \tag{5}$$

$$\mathcal{L}_{Self} = \mathbb{E}_{\boldsymbol{x} \sim \mathcal{U}_{\mathcal{X}}} \mathbb{E}_{\boldsymbol{a}' \sim \pi_{\tilde{\theta}}(\cdot|\boldsymbol{x})} \mathbb{E}_{t \sim \mathcal{U}_{T_{AP}}}[||\pi_{\tilde{\theta}}(\boldsymbol{a}'|\boldsymbol{x}) - \pi_\theta(t(\boldsymbol{a}')|\boldsymbol{x})||_1] \tag{6}$$

**Expert Exploitation.** The major role of expert exploitation is to train high-quality symmetric contextualized policy for various task-conditions $x$ by leveraging the offline expert data $\boldsymbol{a}^*$ with $T_{AP}$. The $T_{AP}$ transforms the existing offline expert data $\boldsymbol{a}^*$ for $P$ times to *augment* the offline expert dataset $D_{exp} = \{(x^{(i)}, a^{(i)*})\}_{i=1}^N$ to reflect the AP-symmetric nature of the placement task. Specifically, we randomly choose $\{t_1, ..., t_P\} \subset T_{AP}$ to generate $D_{aug} = \{(x^{(i)}, a^{(i)*}), (x^{(i)}, t_1(a^{(i)*})), ..., (x^{(i)}, t_P(a^{(i)*}))\}_{i=1}^N$. Then, $\mathcal{L}_{Expert}$ is expressed as a *teacher-forcing* imitation learning scheme with the augmented expert dataset $D_{aug}$. Note that expert exploitation is expected to reduce order bias defined with the three uniform distributions; $\boldsymbol{x}$ of $\mathcal{U}_{D_{exp}}(\boldsymbol{x})$, $\boldsymbol{a}$ of $\mathcal{U}_{D_{exp}}(\boldsymbol{a})$, and $t$ of $\mathcal{U}_{T_{AP}(t)}$.

**Self-Exploitation.** While $D_{aug}$ only contains expert quality, self-exploitation involves self-generated data, whose quality is poor at the beginning but improves over the phase of training. Thus, the self-exploitation scheme is designed to induce the AP-symmetricity in a wider action space to achieve greater generalization capability. Formally, self-exploitation loss is a special form of order bias defined based on the distributions, $\boldsymbol{x} \sim \mathrm{U}_{\mathcal{X}}$, $\boldsymbol{a} \sim \pi_{\tilde{\theta}}$ (current policy) and $t \sim \mathrm{U}_{T_{AP}}$, where $\mathcal{U}$ is a uniform distribution; $\mathcal{L}_{Self} = b(\pi_\theta, \boldsymbol{p} = \{\mathcal{U}_{\mathcal{X}}, \pi_{\tilde{\theta}}, \mathcal{U}_{T_{AP}}\})$.

### 3.3 CONTEXTUAL ATTENTION MODEL

To further improve the generalization capability of the trained DPP solver, we modified the attention model (AM) (Kool et al., 2019) and termed *contextual attention model*. As described in Fig. 2, the decision-making procedure consists of two newly devised context neural networks; (1) encoder capturing initial design conditions while contextualizing the probing port through the probing port context network (PCN), and (2) decoder sequentially allocating decaps on PDN while contextualizing the stages of the partial solution through the recurrent context network (RCN). See Appendix C for detailed implementation.

**Encoder.** Our encoder consists of multi-head attention (MHA) and feedforward (FF), similar to the transformer network (Vaswani et al., 2017). The encoder takes the task-conditioned PDN feature vector $x$ as an input and outputs all node embedding $h$. Encoding is processed once at the initial state, $t = 0$. The node embedding $h$ is time-invariant (i.e., fixed after the encoding process) and is used in the decoding process. We proposed a novel probing context network (PCN) in the encoding process so that the learned solver can adapt well to a new task. PCN is a simple but effective two-layer perceptron model with a ReLU activation layer that takes the hidden embedding of the probing port node $h_{probe}$ and outputs the probing port contextual vector $c_{probe} = \mathbf{MLP}_{PCN}(h_{probe})$.

**Decoder.** With the node embedding $h$ generated by the encoder, decoder sequentially selects an action $a_t$ until placing all $K$ decaps. At each step $t$, the decoder takes (1) the node embedding $h$ (static information), (2) the current state $s_t$, (3) the probing port contextual vector $c_{probe}$, and (4) the previous action context vector $c_{a_{t-1}} = \mathbf{MLP}_{RCN}(h_{a_{t-1}})$ generated by the recurrent context network (RCN) as inputs and outputs a new action $a_t$. To leverage sequential state transitions in decoder, we devised the recurrent context network (RCN). RCN is a two-layer perceptron model with a ReLU activation layer that embeds the previously selected node's embedding $h_{a_{t-1}}$ at step $t$ into $c_{a_{t-1}}$. Then, overall context vector is updated as $c = c_{probe} + c_{a_{t-1}}$ and is used as query for attention decoder that eventually infers the next action $a_t$ by the attention mechanism, where the key and value comes from $h$. See Appendix C.2 for detailed process.

## 4 RELATED WORKS

**Machine Learning-based Methods for DPP.** Deep reinforcement learning (DRL) has been widely used to solve DPP. Park et al. (2018) employed Q-learning and Park et al. (2020); Zhang et al. (2020) applied convolutional neural network (CNN)-based Q approximators to solve a target DPP. However, their methods require large iterations involving costly reward calculations and their trained policies were non-reusable; if the DPP condition changes, they must be re-trained. In an effort to overcome the reusability limitation by training a solver, Park et al. (2022); Kim et al. (2021) implemented promising neural combinatorial optimization (NCO) models, the attention model (Kool et al., 2019) and pointer network (Vinyals et al., 2015), to construct a contextualized policy without iterative exploration and domain knowledge. However, their methods still showed poor data-efficiency in training and unsatisfactory generalization performance.

**Symmetricity Learning in Solution Space.** There exists several studies to leverage the symmetricity in solution space. Kwon et al. (2020) suggested a new reinforcement learning scheme, a policy optimization for multiple optima (POMO) to leverage the traveling salesman problem (TSP)'s solution symmetricity, the cyclic property that identical solution can be expressed as $N$ heterogeneous trajectories by permuting initially visited node. Kim et al. (2022) proposed the symmetric neural combinatorial optimization (Sym-NCO) method, which is an extension of POMO to general-purpose symmetric learning for various combinatorial optimization tasks. Bengio et al. (2021) proposed a generative flow net (GFlowNet) to train policy distribution proportional to reward distribution $\pi \propto R$ considering solution symmetricity. The GFlownet suggests a sequential decision-making scheme with a directed acyclic graph (DAG), instead of classical tree structure, to induce solution symmetricity. The GFlowNet is applied to solve molecule optimization and bio-sequential design (Jain et al., 2022).

While POMO (Kwon et al., 2020) and Sym-NCO Kim et al. (2022) leverage DRL, CSE focuses on offline imitation learning. Though GFlowNet (Bengio et al., 2021) can be trained in a fully offline manner, it is not yet designed for training a contextualized policy. Thus, CSE is positioned between POMO and Gflownet as an offline symmetricity learning method to train contextualized policy.

Table 1: Performance evaluation with the average score of 100 PDN cases (the higher the better).

| Method | Method Type | PI Simulation ($M$) | Avg. Score |
|---|---|---|---|
| Random Search | Online Search | 10,000 | 12.70 |
| Genetic Algorithm (*expert policy*) | Online Search | 100 | 12.56 |
| Genetic Algorithm | Online Search | 500 | 12.79 |
| AM-RL (Park et al., 2022) | Pretrained | 1 | 11.71 |
| Arb-RL (Kim et al., 2021) | Pretrained | 1 | 9.60 |
| AM (Park et al., 2022)-IL | Pretrained | 1 | 12.06 |
| Arb (Kim et al., 2021)-IL | Pretrained | 1 | 10.80 |
| **CSE** (*ours*) | Pretrained | 1 | **12.88** |

## 5 EXPERIMENTAL RESULTS

### 5.1 SETUP

**PDN and Decap Specifications.** The PDN used for verification is a chip-package hierarchical PDN, modeled by the segmentation method (Kim et al., 2010; Cho et al., 2019). The PDN model is represented as $(N_{row} \times N_{col}) = (10 \times 10)$ grids over 201 frequency points linearly distributed between 200MHz and 20GHz, which gives $\mathbf{100 \times 100 \times 201 \approx 2M}$ impedances to be evaluated per each task; the PDN scale is reasonable to reflect the simulation intensiveness of DPP. The RLGC electrical parameters of PDN unit-cell are shown in Appendix A.2. Out of the $N_{row} \times N_{col}$ ports on PDN, one is assigned as a probing port and 0 to 15 ports are assigned as keep-out ports (see Appendix A.4). Decap is modeled as a unit-cell with a single port that is attached to a specific port on PDN when an action is made. The RLGC electrical parameters of decap unit-cell are also shown in Appendix A.2. 100 PDN cases for test and another 100 PDN cases for validation were generated for performance evaluation. We made sure test data, validation data and training data did not overlap. We used the number of decap $K = 20$ for every training, but $K$ can be changed during the inference.

**Offline Expert Data Collection.** Since expert data for the DPP benchmark was not available, we synthetically generated offline expert data using genetic algorithm (GA) for this study. The number of iterations done for collecting a single data is represented as $M$. Note that the higher the $M$, the better the quality of data, but the higher the simulation cost. We used GA$\{M = 100\}$ to collect the offline expert data. In addition, we denote $N$ as the number of offline expert data used for training CSE. Note that the lower the $N$ the more data-efficient the training is.

**Hyperparameters.** For training, we generate three transformed data per expert data. We denote $P(= 3)$ as the number of AP-transformed data per offline data. Thus, the total number of guiding data becomes $N \times (P + 1)$. For instance, $P = 3, N = 50$ makes total $50 \times 3 + 50 = 200$ guiding data. We set the distribution $\rho$, described in Section 2.1, as uniform distribution for training.

We used $N = 2000$ offline expert data for training CSE and IL-based baselines. For the learning algorithm, we used ADAM (Kingma & Ba, 2015) with a learning rate of $10^{-5}$. We trained our model with batch size 100 for $N < 200$ and batch size $1,000$ for $N = 1,000$ and $2,000$. We trained for a maximum of 200 epochs for each model; we used the model with the best validation score for CSE and other ML baselines to evaluate performance. See Appendix D for detailed setup.

**Baselines for Comparison.** For search heuristic baseline methods, we implemented random search and genetic algorithm. Since these methods are iterative solvers, they require a large number of simulations (i.e., $M >= 100$. See Table 1) for each problem. For non-iterative learning-based solvers, we reported two RL baselines, AM-RL (Park et al., 2022) and Arb-RL (Kim et al., 2021) and two IL baselines, AM-IL and Arb-IL, which are modified AM-RL and Arb-RL with imitation learning instead of reinforcement learning, to investigate the effectiveness of CSE components compared to the same imitation learning approaches with different learning strategies/neural architecture. Implementation details of the baselines are provided in Appendix D.2 and Appendix D.3.

### 5.2 GENERALIZATION CAPABILITY EVALUATION

To verify the generalization capability of the trained solver, each method is given the same unseen 100 DPPs and the average performance score was measured, after allocating $K = 20$ decaps on each.

As shown in Table 1, our CSE significantly outperformed all baselines in terms of average performance score. Online search methods generally find solutions that give a high average performance. This is due to a large number of searching iterations $M$, which incurs the same number of costly simulations. On the other hand, the learning-based baselines and CSE do not require simulations to generate solutions; once trained, they only require a single simulation to measure the performance. Though learning-based methods can easily find such a solution, CSE is the only method capable of finding a solution that outperforms the highly iterative online search methods by a zero-shot inference.

When the number of costly simulations was limited, RL-based methods (AM-RL, Arb-RL) showed poorer generalization capability than their IL versions (AM-IL, Arb-IL) due to inefficiency in exploring over extremely large combinatorial action space of DPP. We believe that imitation learning approach, fitting the policy with offline expert data, has greater exploration capability with the help of expert policy thus able to achieve higher performance with a limited simulation budget (see Appendix D.2). Note that if we have an infinite budget for reward simulation (which never happens in a real-world hardware setting), DRL could achieve greater performance and generalization capability with a sufficient learning loop.

Among the IL approaches trained with the same number of offline expert data ($N = 2,000$), CSE showed the highest performance. We believe that such higher generalization capability comes from both symmetricity exploitation schemes and the newly devised neural architecture: (1) expert exploitation and self-exploitation with symmetric label transformation amplify the number of data to train with and induce solution symmetricity to improve generalization capability. (2) the neural architecture with PCN and RCN makes the policy easily adapt to new task conditions.

**Extrapolation over Expert Method.** The CSE policy trained with offline expert data generated by the expert policy, GA{100}, outperformed GA{500}, with zero-shot inference. That is, the CSE policy trained with low-quality offline expert data produced higher-quality designs. We believe this was possible because we trained a factorized form of policy that does not predict labels in a single step but produced a solution through a serial iterative roll-out process, during which a good strategy for placing decaps can be identified. In addition, the CSE with symmetric label transformation has further guided the policy to learn such an effective decap placement design scheme.

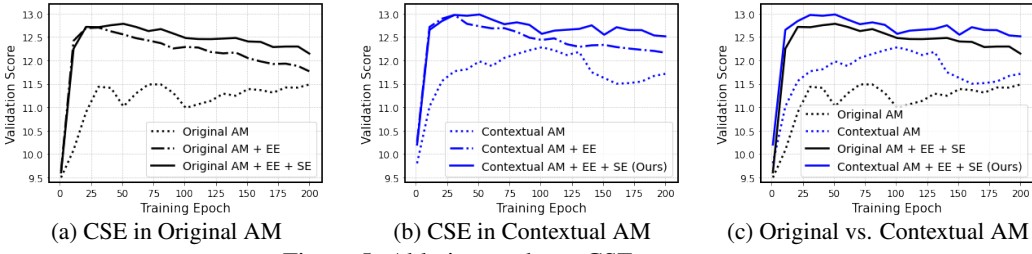

(a) CSE in Original AM  (b) CSE in Contextual AM  (c) Original vs. Contextual AM

Figure 5: Ablation study on CSE components

**Ablation Study.** We conducted ablation studies to validate the effectiveness of CSE components and context neural networks with sparse offline data ($N = 100$). We ablated the effectiveness of expert exploitation (EE) and self-exploitation (SE) in two policy networks: original AM and contextual AM (ours). The original AM refers to the AM-IL baseline. Each component of CSE supported increasing generalization capability in both policy networks and the contextual AM with newly devised context neural networks was verified to outperform the original AM. Therefore, we verified that both CSE components and the modifications of AM successfully contributed to the promising performance.

**Order Bias Measurement.** We empirically show that our CSE successfully induces AP-symmetricity by reducing the order bias (see Appendix H).

## 5.3 OFFLINE DATA EFFICIENCY EVALUATION

We investigated how the number $N$ of offline data generated by the expert method, GA{$M = 100$}, affects the design performance of CSE and the two baselines, AM-IL and Arb-IL. As shown in Fig. 6, CSE outperformed the baselines in all $N$ variation; CSE trained with $N = 100$ even performed better than the baselines trained with $N = 2000$. Moreover, CSE monotonically improved with $N$ while the others saturated when $N > 500$.

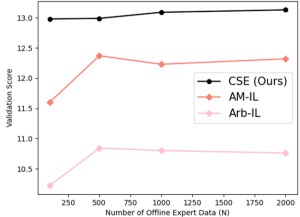

Figure 6: Offline data-efficiency evaluation ($P = 3, \lambda = 8$)

Table 2: Scalability evaluations on larger PDN scale and varying number of decap $K$.

| Scale Variables | | Methods | | |
|---|---|---|---|---|
| PDN Scale | Number of Decap, $K$ | GA {100} | AM (Park et al., 2022)+IL | **CSE (ours)** |
| | 12 | 11.77 | 10.22 | **12.23** |
| | 16 | 12.25 | 11.13 | **12.60** |
| $10\times10$ | 20 | 12.53 | 11.71 | **12.81** |
| | 24 | 12.79 | 12.20 | **12.95** |
| | 30 | 13.02 | 12.62 | **13.11** |
| $15\times15$ | 20 | 7.61 | 6.23 | **8.47** |
| | 40 | 7.69 | 7.75 | **8.54** |

## 5.4 SCALABILITY EVALUATION

For scalability verification, learning-based DPP methods were pre-trained for a fixed scale PDN, $(10 \times 10)$, and a fixed number of decaps, $K = 20$. Then, the pre-trained models were asked to place decaps of varying $K \in \{12, 16, 20, 24, 30\}$ on $(10 \times 10)$ PDN and varying $K \in \{20, 40\}$ for a larger $(15 \times 15)$ PDN without additional training (i.e, zero-shot). We chose two baseline methods for comparison: GA {100} and AM-IL. As shown in Table 2, our CSE outperformed GA {100} and AM-IL for all scales. Furthermore, CSE achieved greater performance with fewer decaps. Reducing the number of decaps has a significant industrial impact; as hardware devices are mass-produced, reducing a single decap saves enormous fabrication cost.

## 5.5 FLEXIBILITY VERIFICATION FOR PRACTICAL APPLICATION

To verify the practical applicability, we applied the proposed CSE to a real-world hardware application, high bandwidth memory (HBM), which is an interposer-based 2.5D IC. As shown in Fig. 7, the hierarchical PDN model of HBM is composed of $(40 \times 40)$ package PDN, $(40 \times 60)$ interposer PDN and $(15 \times 20)$ on-chip PDN, each layers connected by TSV + C4 bumps and microbumps.

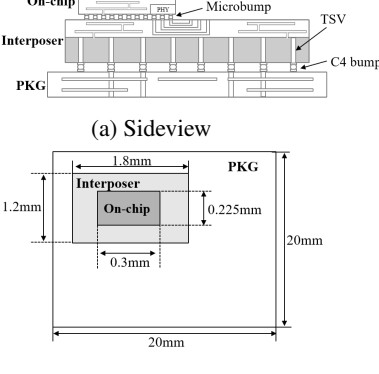

(a) Sideview

(b) Topview

Figure 7: Structure of HBM PDN model

For performance evaluation, we compared the CSE$\{N = 1000\}$ to GA$\{M = 100\}$, AM-IL$\{N = 1000\}$ and Arb-IL$\{N = 1000\}$ on placing varying number of decaps, $K$, for 100 test cases. The pre-trained solvers with $K = 20$ were used for CSE, AM-IL and Arb-IL without additional training.

Fig. 8 verifies that CSE achieved higher performance with significantly fewer decaps. For instance, the performance score of 26.68, attained with 40 decaps by GA$\{M = 100\}$ and 34 decaps by AM-IL, was achieved with only 26 decaps by zero-shot CSE. Arb-IL was unable to achieve 26.68 even with 80 decaps. Power noise analysis in a test case was carried out (see Appendix E.3), where the initial power noise before placing decap was 10.546mV. CSE was able to reduce the power noise to 0.610mV (-94.2%) with 26 decaps while GA$\{M = 100\}$ reduced to 0.682mV (-93.5%) with 40 decaps. Reducing the number of decaps is a huge contribution to the industry as hardware devices are mass-produced, reducing a single decap can greatly reduce production costs.

## 6 CONCLUSION

This paper proposed the *collaborative symmetricity exploitation* (CSE) framework for training contextualized policy (i.e., solver) of placement tasks in an offline manner. The CSE was applied to decap placement problem (DPP) and achieved the most promising performance among all baseline methods. The CSE was also validated on several scales and on multiple hardware devices. The CSE is a general purpose offline learning scheme for placement tasks that can be further applied to other hardware placement tasks including chip placement, ball grid array (BGA) placement, and via placement.

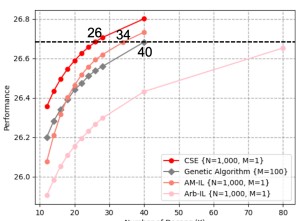

Figure 8: Performance comparison with number of decap variation on HBM

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

## A   DPP Electrical Modeling and Problem Definition

This section provides electrical modeling details of PDN and decap models used for verification of CSE in DPP. Note that these electrical models can be substituted by those of your interest. There are three methods to extract PDN and decap models that are also used for objective evaluation; 3D EM simulation tool, ADS circuit simulation tool, and unit-cell segmentation method. For each method, there exists a trade-off between time complexity and accuracy. See Table 3. Out of the three methods, we used the unit-cell segmentation method for a benchmark. Simulation time was evaluated using the same PDN model on Intel i7. Note that simulation time depends on the size and complexity of the PDN model.

Table 3: Time Taken for an Objective Evaluation of a PDN model described in Appendix A.2

| Simulation Method | Time Taken |
|---|---|
| EM Simulation Tool | ≈10 hours |
| ADS Circuit Simulation Tool | 23.58 sec |

### A.1   Domain Perspective Decap Placement Problem

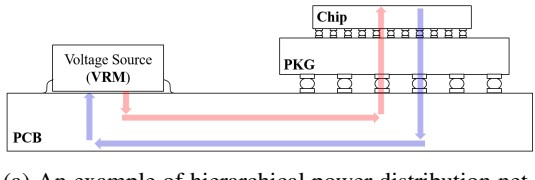

(a) An example of hierarchical power distribution network (PDN).

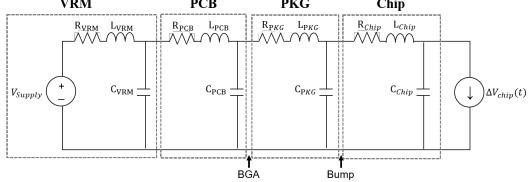

(b) Electrical circuit model of the hierarchical PDN in (a).

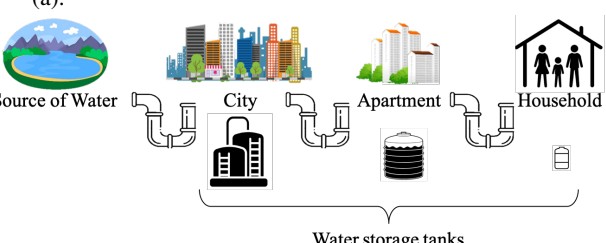

(c) Water supply chain from the source to household.

Figure 9: Illustration of Hierarchical Power Distribution Network (PDN) analogous to Water Supply Chain.

The development of AI has led to an increased demand for high-performance computing systems. High-performance computing systems not only require precise design of hardware chips such as CPU, GPU and DRAM, but also require stable delivery of power to the operating integrated circuits. Power delivery has become a huge technical bottleneck of hardware devices due to the continuously decreasing supply voltage margin along with the technology shrink of CMOS transistors (Hwang et al., 2021).

Fig. 9 (a) shows the power distribution network (PDN) consisting of all the power/ground planes from the voltage source to operating chips. Power is generated in VRM and delivered through

electrical interconnections of PCB, package and chip. Finding ways to meet the desired voltage and current from the power source to destinations along the PDN is detrimental because failure in achieving power integrity (PI) leads to various reliability problems such as incorrect switching of transistors, crosstalk from neighboring signals, and timing margin errors (Swaminathan & Engin, 2007). Decoupling capacitors (decaps) placed on the PDN allows the reliable power supply to the operating chips, thus improving the power integrity of hardware. As shown in Fig. 9 (b)-(c), the role of decap is analogous to that of water storage tanks, placed along the city, apartment, and household, that can provide water uninterruptedly and reliably. As if placing more water tanks can make the water supply more stable, placing more decaps can make power supply more reliable. However, because adding more decaps requires more space and is costly, optimally placement of decaps is important in terms of PI and cost/space-saving.

### A.2 PDN and Decap Models for Verification

**Unit-Cell Segmentation Method.** The segmentation method (Kim et al., 2010) is a simple and fast way to generate approximated electrical models. Because the analysis of the full electrical model using EM simulation is very time-consuming, we divided the full PDN model into smaller unit-cells and constructed the full PDN model using the unit-cell segmentation method. For fast simulation, we used equation-based python implemented segmentation method, illustrated in Fig. 10.

Segmentation method was used for generation of PDN model consisting of a chip layer and a package layer for verification as illustrated in Fig. 10 (a). The segmentation method was also used for objective evaluation of DPP. When a solution for DPP is made, decaps are placed on the corresponding ports on PDN using the segmentation method as illustrated in Fig. 10 (b).

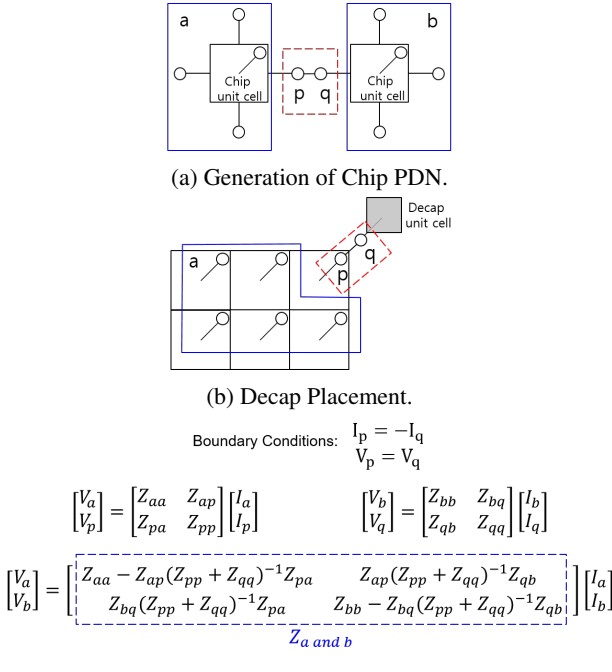

(a) Generation of Chip PDN.

(b) Decap Placement.

Boundary Conditions: $I_p = -I_q$
$V_p = V_q$

$$\begin{bmatrix} V_a \\ V_p \end{bmatrix} = \begin{bmatrix} Z_{aa} & Z_{ap} \\ Z_{pa} & Z_{pp} \end{bmatrix} \begin{bmatrix} I_a \\ I_p \end{bmatrix} \qquad \begin{bmatrix} V_b \\ V_q \end{bmatrix} = \begin{bmatrix} Z_{bb} & Z_{bq} \\ Z_{qb} & Z_{qq} \end{bmatrix} \begin{bmatrix} I_b \\ I_q \end{bmatrix}$$

$$\begin{bmatrix} V_a \\ V_b \end{bmatrix} = \underbrace{\begin{bmatrix} Z_{aa} - Z_{ap}(Z_{pp} + Z_{qq})^{-1}Z_{pa} & Z_{ap}(Z_{pp} + Z_{qq})^{-1}Z_{qb} \\ Z_{bq}(Z_{pp} + Z_{qq})^{-1}Z_{pa} & Z_{bb} - Z_{bq}(Z_{pp} + Z_{qq})^{-1}Z_{qb} \end{bmatrix}}_{Z_{a \text{ and } b}} \begin{bmatrix} I_a \\ I_b \end{bmatrix}$$

(c) Segmentation Method.

Figure 10: Segmentation Method Implemented for PDN Generation and Decap Placement on PDN.

The PDN model we used for verification has a two-layer structure; a package layer at the bottom and a chip layer on top of it as illustrated in Fig. 11. The PDN was modeled through the unit-cell segmentation method. Package layer was composed of $40 \times 40$ package unit-cells and chip layer was composed of $10 \times 10$ (i.e, $N_{row} \times N_{col}$) chip unit-cells. Because the DPP benchmark places MOS type decaps, which are placed on chip, ports are only available on chip. Thus, we extracted $10 \times 10$ ports information from the chip layer. See Fig. 14 (a), illustrating the chip PDN divided into $10 \times 10$ units and each unit-cell numbered.

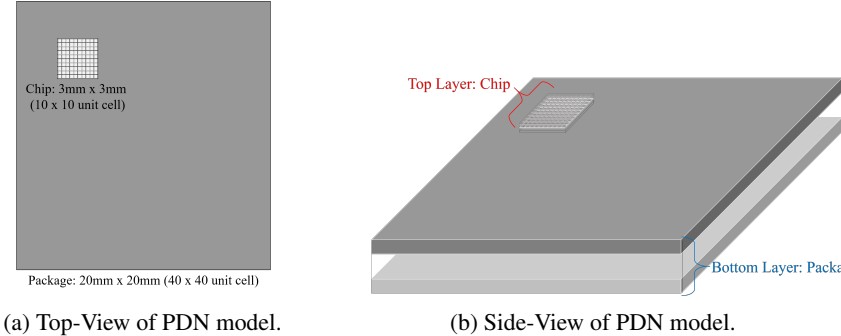

(a) Top-View of PDN model.

(b) Side-View of PDN model.

Figure 11: Top-view and Side-view of PDN Model used for Verification

The electrical models of package and chip unit-cells that are used to build the PDN model for verification are described in Fig. 12. The chip layer is composed of $10 \times 10$ unit-cells, and the package layer is composed of $40 \times 40$ unit-cells using the segmentation method. The corresponding values of electrical parameters are listed in Table 4.

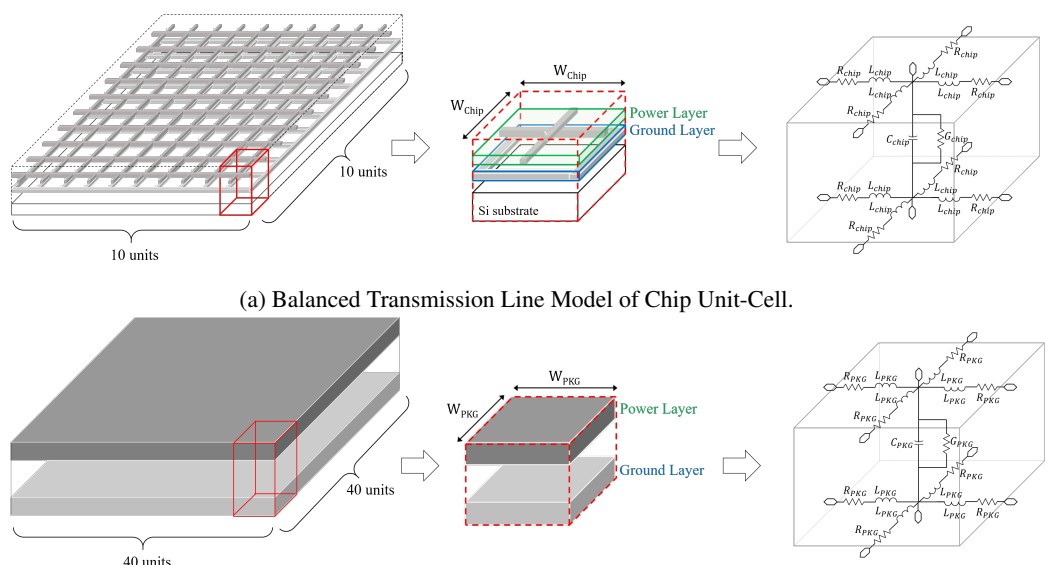

(a) Balanced Transmission Line Model of Chip Unit-Cell.

(b) Balanced Transmission Line Model of Package Unit-Cell.

Figure 12: Electrical Modeling of Chip and Package Unit-Cells for PDN Model generation.

Table 4: Width and Electrical Parameters for Chip and Package Unit-Cells used for Verification

| Unit-Cell Model | W | R | L | G | C |
|---|---|---|---|---|---|
| Chip | $300\mu$m | $0.26\,\Omega$ | 22pH | 1.2mS | 0.77pF |
| Package | 0.5mm | $0.093\,\Omega$ | 0.25nH | $5.4\mu$S | 0.045pF |

We implemented MOS type decap for verification. Decap model and its electrical parameters are shown in Fig. 13. As mentioned in appendix A.1 Fig. 10 (b), the solution to DPP is evaluated using the segmentation method.

Note that these electrical parameters and PDN structures were used as a benchmark. For practical use of CSE, these PDN and decap models can be substituted by those of your interests.

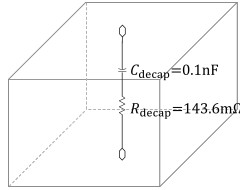

Figure 13: Decap Unit-Cell with the Electrical Parameters used for Verification.

## A.3 INPUT PROBLEM PDN AND OUTPUT DECAP PLACEMENT DATA STRUCTURE

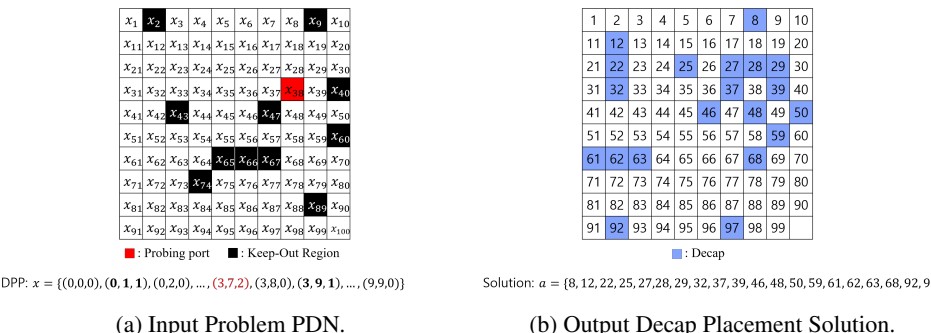

(a) Input Problem PDN.

(b) Output Decap Placement Solution.

Figure 14: Illustration of how the DPP problem with specific condition is given as an input and decap placement solution is generated as an output.

Each unit-cell (i.e, port) of the PDN model described in Appendix A.2 is represented as a set of 3D feature vectors composed of x-coordinate, y-coordinate and port condition; 1 representing keep-out region, 2 representing a probing port and 0 for the decap allowed ports. Total $10 \times 10$ (i.e, $N_{row} \times N_{col}$) 3D vectors represent the problem PDN. The solution to DPP is the placement of decaps. As illustrated in Fig. 14 (b), the solution is given as a set of port numbers corresponding to each decap location.

## A.4 RANDOM PROBLEM GENERATION OF DPP

To randomly generate decap placement problems (DPPs) with distinct conditions for training, test and validation, a probing index $I_{probe}$ is selected randomly from a uniform distribution of $\{1, ..., N_{row} \times N_{col}\}$. Then keep-out region indices $I_{keepout}$ are randomly selected through the following two stages: the number of keep-out regions $|I_{keepout}|$ is randomly selected from a uniform distribution of $0 \sim 15$. Then, a set of indices of keep-out ports $I_{keepout}$ is generated by random selection from the uniform distribution of $\{1, ..., N_{row} \times N_{col}\}$. We generated 100 test problems and 100 validation problems for $10 \times 10$ PDN and 50 test problems and 50 validation problems for $15 \times 15$ PDN. We made sure the training, test, and validation problems do not overlap.

## B EXPERT LABEL COLLECTION

We used a genetic algorithm (GA) as the expert policy to collect expert guiding labels for imitation learning. GA is the most widely used search heuristic method for DPP (Erdin & Achar, 2019; de Paulis et al., 2020; Xu et al., 2021; Juang et al., 2021). We devised our own GA for DPP, the objective of which is to find the placement of given number ($K$) of decaps on PDN with a probing port and 0-15 keep-out regions that best suppresses the impedance of the probing port.

**Notations.** $M$ is the number of samples to undergo an objective evaluation to give the best solution. The value of $M$ is defined by the size of population $P_0$ times the number of generation $G$. $K$ refers to the number of decaps to be placed. $P_{elite}$ is the number of elite population.

**Guiding Dataset.** To generate expert labels, guiding problems were generated in the same way test dataset was generated. We made sure the guiding data problems do not overlap with the test dataset problems. Also, we made sure each guiding problem does not overlap with each other. Each guiding data problem goes through the following process described in Fig. 15 to collect the corresponding expert label.

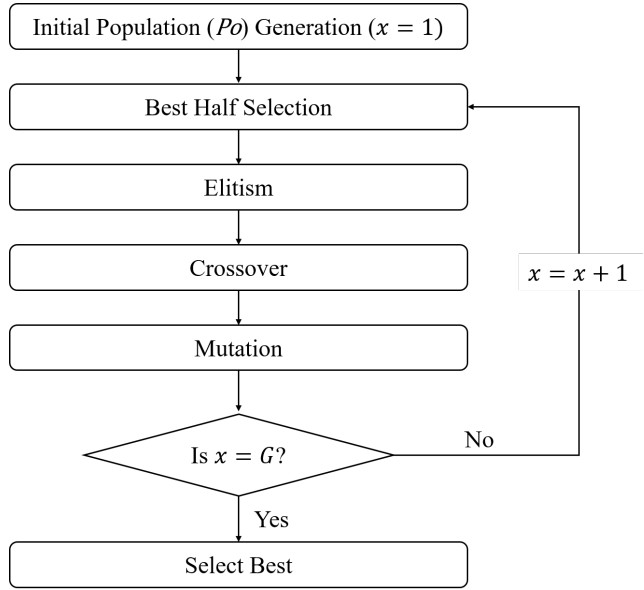

Figure 15: Process Flow of Genetic Algorithm for DPP.

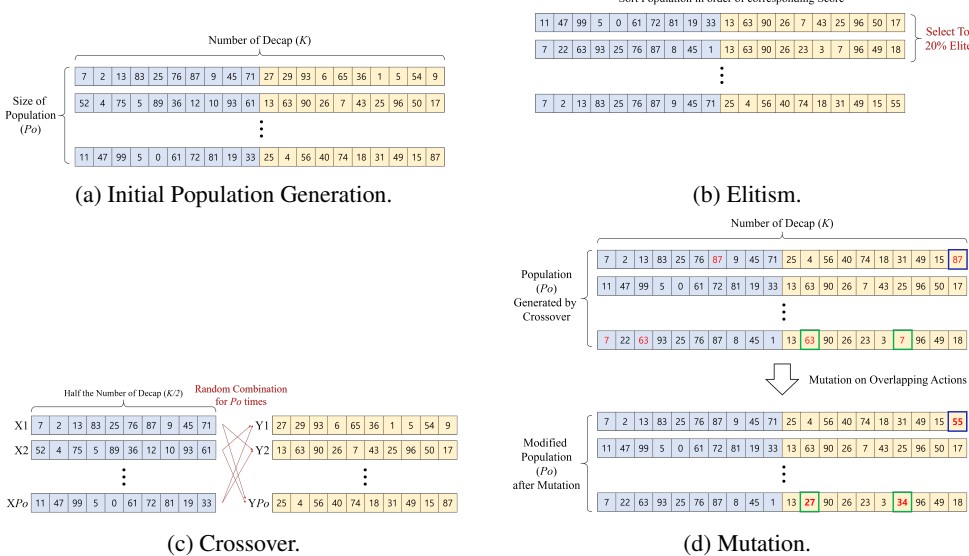

Figure 16: Illustration of each GA Operators used for DPP Guiding Data Generation.

**Population and Generation.** For GA $\{M = 100\}$ (*expert policy*), we fixed the size of population as $P_0 = 20$ and the number of generation as $G = 5$, which makes up total number of samples to be $M = P_0 \times G = 100$. Each solution in the initial population is generated randomly. As described in Fig. 14 (b), each solution consists of $K$ numbers, each representing a decap location on PDN. Note that each solution consists of random numbers from 0 to 99 except numbers corresponding to probing port and keep-out region locations.

Once the initial population is generated randomly, a new population is generated through elitism, crossover, and mutation. This whole process of generating a new population makes one generation; the Generation process is iterated for $G - 1$ times.

**Elitism.** Once initial population is formulated, the entire population undergoes objective evaluation and gets sorted in order of objective value. The size of elite population is pre-defined as $P_{elite} = 4$ for GA $\{M = 100\}$ (*expert policy*). That means the top 4 solutions in the population become the elite population and are kept for the next generation.

**Crossover.** Crossover is a process by which new population candidates are generated. Each solution of the current population including the elites is divided in half. Then, as described in Fig. 16 (c), half the solutions on the left and the other half on the right go through random crossover for $P_0$ times to generate a new population. If the elite population is available, $P_0 - P_{elite}$ random crossover takes place so that the total population size becomes $P_0$, including the elite population.

**Mutation.** According to Fig. 16 (d), there may exist solutions with overlapping numbers after the random crossover. We replace the overlapping number with a randomly generated number, and we call this mutation.

**Select Best.** When $G$ is reached, the final population is evaluated by the performance metric. Then, a solution with the highest objective value becomes the final guiding solution for the given DPP.

The guiding problems and corresponding solutions generated as a result of GA are saved and used as guiding expert labels for imitation learning.

## C  DETAILS OF NEURAL ARCHITECTURE DESIGN

Our neural architecture has the AM (Kool et al., 2019) with context modification. The AM is a transformer(Vaswani et al., 2017)-based encoder-decoder model designed to solve combinatorial optimization problems. We used conventional notations from transformer (Vaswani et al., 2017) and AM (Kool et al., 2019), including multi-head attention (MHA), feed forward (FF), query, key and value ($Q, K, V$). Because their terminologies are well organized, we tried to keep every notation as possible. In this paper, we focused on presenting the main differences between AM and our architecture. See Kool et al. (2019) for detailed mechanism of AM.

### C.1  CHANGE OF NOTATIONS.

There are small revisions we made from Kool et al. (2019). In AM, TSP nodes are presented as $\boldsymbol{x}_i$, $i \in \{1, ..., N\}$, where $N$ refers to the number of TSP nodes. This paper uses $I_{probe}$ for the node of the probing port, $I_{keepout}$ for nodes of the keep-out regions and $I_{allowed}$ for nodes of the decap-allowed ports.

Kool et al. (2019) denotes action as $\boldsymbol{\pi}$ (for representing permutation action), but we denoted action as $\boldsymbol{a}$.

In, Kool et al. (2019), the notation, $\boldsymbol{h}^{(N)}$, refers to $N$ times MHA in encoder; we denoted this notation as $\boldsymbol{h}$ just for readability.

There are two additional notations: $\boldsymbol{c}_{probe}$ is the probing context embedding from the probing port context network (PCN in section 3.2) and $\boldsymbol{c}_{a_{t-1}}$ is the recurrent context embedding from the recurrent context network (RCN in section 3.2) for $step = t$.

### C.2  HIGHLIGHT OF MODIFICATIONS: CONTEXT EMBEDDING.

The main difference between the AM and ours is the context embedding and is illustrated in Fig. 17.

AM's (Kool et al., 2019) context embedding is presented as follows:

$$\boldsymbol{h_{(c)}} = MHA([\boldsymbol{h_{(g)}}, h_{a_{\tau-1}}, \boldsymbol{h_{a_1}}], \boldsymbol{h}) \tag{7}$$

**Context embedding of AM.** Since the AM was originally designed for TSP and its invariant problems, AM's context embedding is implemented for capturing the entire graph by taking the average of all

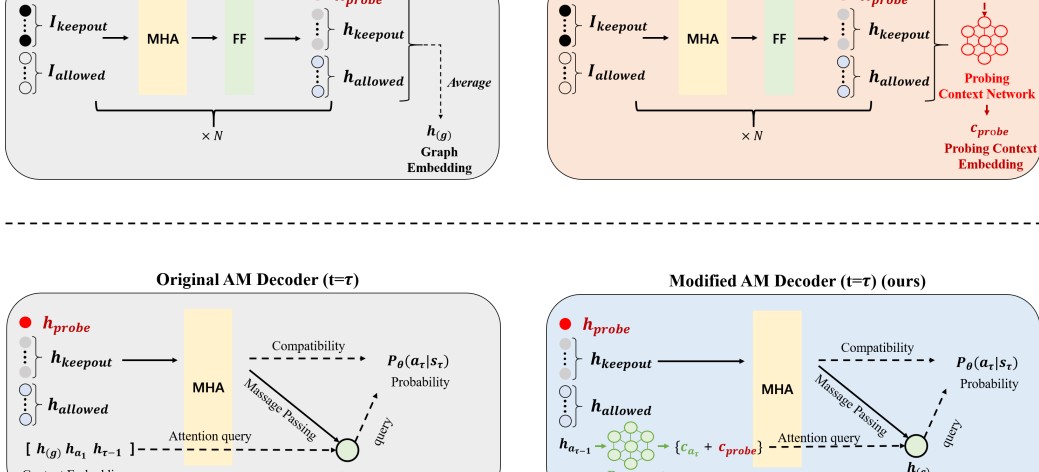

Figure 17: Overview of main difference between AM and modified version of AM.

node embedding, $h_{(g)}$, state-transition with $h_{a_{\tau-1}}$ and final destination with $h_{a_1}$. Note that TSP is a routing problem, where it must return to the first node (i.e, destination node is first visited node).

**Context embedding of AM for DPP (AM-RL (Park et al., 2022)).** Park et al. (2022) also used the AM for decap placement with modification of context embedding. Park et al. (2022) tried to add $h_{probe}$ to capture the location of probing port as follows:

$$h_{(c)} = MHA([h_{(g)}, h_{a_{t-1}}, h_p], h) \tag{8}$$

**Context embedding of Ours.** We observed that $h_{(g)}$ degrades the performance of the model for DPP. DPP is different from TSP; we need a new DPP-specific context embedding strategy. Therefore, we tried to focus on the probing port more than others by proposing the PCN. We removed $h_{(g)}$ and $h_{a_1}$ from the context embedding and replaced them with our newly designed context embedding. Our context embedding is described as follows:

$$h_{(c)} = MHA(c_{probe} + c_{a_{t-1}}, h) \tag{9}$$
$$c_{probe} = \mathbf{MLP}_{PCN}(h_{probe}) \tag{10}$$
$$c_{a_{t-1}} = \mathbf{MLP}_{RCN}(h_{a_{t-1}}) \tag{11}$$

Note that both $\mathbf{MLP}_{PCN}$ and $\mathbf{MLP}_{RCN}$ are two-layer perceptron models with ReLU activation, where input and output dimensions are identical ($d = 128$ in all experiments).

## C.3 CALCULATION OF PROBABILITY.

Probability calculations using context hidden embedding $h_{(c)}$, and PDN hidden embedding $h_i$, $i \in \{1, ..., N_{row} \times N_{col}\}$ in (11-14) are exactly identical to (5-8) in Kool et al. (2019) except the masking mechanism in equation 13 and equation 14. Because Kool et al. (2019) solves TSP, so they mask the previously selected actions by forcing $-\infty$ as compatibility $u_{(c)j}$. For DPP, we mask not only the previously selected actions $a_{1:t-1}$ but also the probing port index $I_{probe}$ and the keep-out region indices $I_{keepout}$; it is forbidden to choose the $I_{probe}$, $I_{keepout}$ and previously selected actions $a_{1:t-1}$

Query, key and value are computed by:

$$q_c = W^Q h_{(c)}, k_i = W^K h_i, v_i = W^V h_i \tag{12}$$

Note that $W^Q$, $W^K$ and $W^V$ are 128-to-128 linear projections.

After that, compatibility $u_{(c)j}$ is computed by the dot product of query and key, with masking mechanism (setting $-\infty$ not to select actions in $\boldsymbol{s}_{t-1}$).

$$
u_{(c)j} = \begin{cases} \frac{\mathbf{q}_{(c)}^T \mathbf{k}_j}{\sqrt{128}} & \text{if } j \notin I_{probe}, I_{keepout}, a_{1:t-1} \\ -\infty & \text{otherwise} \end{cases} \tag{13}
$$

The $tanh$ clipping is done following Bello et al. (2016) and Kool et al. (2019).

$$
u_{(c)j} = \begin{cases} 10 \cdot \tanh\left(\frac{\mathbf{q}_{(c)}^T \mathbf{k}_j}{\sqrt{128}}\right) & \text{if } j \notin I_{probe}, I_{keepout}, a_{1:t-1} \\ -\infty & \text{otherwise.} \end{cases} \tag{14}
$$

Finally, probability can be computed using softmax function as follows:

$$
p_{\boldsymbol{\theta}}\left(a_t = i \mid \boldsymbol{s}_t\right) = \frac{e^{u_{(c)i}}}{\sum_j e^{u_{(c)j}}} \tag{15}
$$

# D    DETAILED EXPERIMENTAL SETTINGS

This section provides detailed experimental settings for main experiments and ablation studies.

## D.1    TRAINING HYPERPARAMETERS.

There are several hyperparameters for training; we tried to fix the hyperparameters as Kool et al. (2019) did for showing their frameworks' practicality. We then provided several ablation studies on each hyperparameter to analyze how each component contributes to performance improvement.

Training hyperparameters are set to be identical to those presented in AM for TSP (Kool et al., 2019) except learning rate, unsupervised regularization rate $\lambda$, the number of expert data $N$, number of action permutation transformed data per expert data $P$ and batch size $B$.

Table 5: Hyperparameter setting for training model.

| Hyperparameter | Value |
|---|---|
| learning rate | 0.00001 |
| $\lambda$ | $8 \times 10^{32}$ |
| $N$ | 1000 |
| $P$ | 3 |
| $B$ | 1000 |

## D.2    IMPLEMENTATION OF ML BASELINES.

There are two main ML baselines, Arb-RL (Kim et al., 2021) and AM-RL (Park et al., 2022).

**Arb-RL.** Arb-RL is a PointerNet-based DPP solver proposed by Kim et al. (2021). However, reproducible source code was not available. Therefore, we implemented the Arb-RL following the implementation of Bello et al. (2016) [1] and paper of Kim et al. (2021). We set the training step $1,600$ with batchsize $B = 100$ that makes total $160,000$ PI simulation.

**Arb-IL.** Arb-IL is an imitation learning version of Arb-RL trained by our training data. We set $N = 2000$, $B = 1000$ for training Arb-IL.

---

[1]https://github.com/pemami4911/neural-combinatorial-rl-pytorch

**AM-RL.** AM-RL is a AM-based DPP solver proposed by Park et al. (2022). We reproduced AM-RL by following implementation of Kool et al. (2019)[2] and paper of Park et al. (2022). We set the training step $2,000$ with batchsize $B = 100$ that makes total $200,000$ PI simulation.

**AM-IL.** AM-IL is an imitation learning version of AM-RL trained by our training data. For experiments in Table 1, we set $N = 2000$ and $B = 1000$ for training. For ablation study, we mainly ablate $N$, when $N = 100$ we set $B = 100$. Here is the training sample complexity (the number of PI simulations during training) of each ML baselines and CSE:

Table 6: Training sample complexity of ML baselines and CSE.

| Methods | The Number of PI simulations for Training |
|---|---|
| Arb-RL | 200,000 |
| AM-RL | 200,000 |
| Arb-IL $\{N = 2000\}$ | 200,000 ($N = 2000$, $M = 100$ from GA expert) |
| AM-IL $\{N = 2000\}$ | 200,000 ($N = 2000$, $M = 100$ from GA expert) |
| **CSE** $\{N = 100\}$ (ours) | 10,000 ($N = 100$, $M = 100$ from GA expert) |
| **CSE** $\{N = 1000\}$ (ours) | 100,000 ($N = 1000$, $M = 100$ from GA expert) |
| **CSE** $\{N = 2000\}$ (ours) | 200,000 ($N = 2000$, $M = 100$ from GA expert) |

During the inference phase, each learned model produces a greedy solution from their policies (i.e., $M = 1$) following (Kool et al., 2019).

---

[2]https://github.com/wouterkool/attention-learn-to-route

### D.3  IMPLEMENTATION OF META-HEURISTIC BASELINES.

**Genetic Algorithm (GA).** GA $\{M = 100\}$ and GA $\{M = 500\}$ are implemented as baselines. For detailed procedures and operators used for GA, see Appendix.B. GA $\{M = 100\}$ is the expert policy used to generate expert data for imitation learning in CSE. For GA $\{M = 100\}$, the size of population, $P_0$, is 20, number of generation, $G$, is 5 and elite population, $P_{elite}$, is 4. For GA $\{M = 500\}$, $P_0$ is 50, $G$ is 10 and $P_{elite}$ is 10.

**Random Search (RS).** The random search method generates $M$ random samples for a given problem and selects the best sample with the highest objective value.

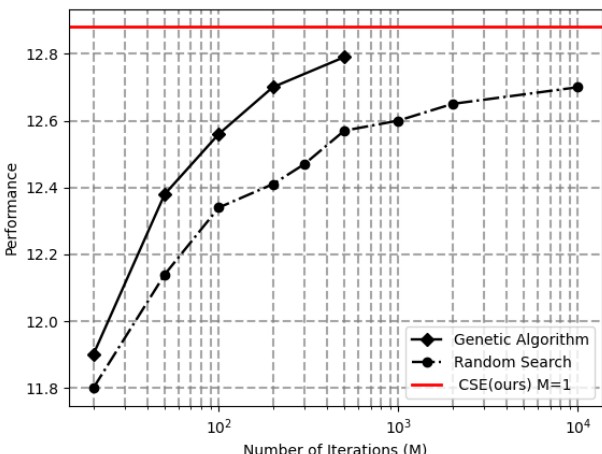

Figure 18: Performance of GA and RS with varying number of iterations ($M$) in comparison to CSE at $M = 1$.

Fig. 18 shows the performance of GA and RS depending on the number of iterations ($M$). The performance was measured by taking the average of 100 test data solved by each method at each $M$. GA outperformed RS at every $M$, and the performance increased with increasing $M$ for both methods. However, the gradient of performance increment decreased with increasing $M$. On the other hand, our CSE showed higher performance than GA$\{M = 100\}$ and RS $\{M = 10,000\}$ with a single inference $M = 1$.

## E  EXPERIMENTAL RESULTS IN TERMS OF POWER INTEGRITY

The objective of DPP is to suppress impedance of the probing port as much as possible over a specified frequency range and is measured by the objective metric, $Obj := \sum_{f \in F}(Z_{initial}(f) - Z_{final}(f)) \cdot \frac{1\text{GHz}}{f}$. Performance of CSE was evaluated in comparison to GA $\{M = 100\}$ (*expert policy*), GA $\{M = 500\}$, RS $\{M = 10,000\}$, AM-RL and AM-IL on unseen 100 PDN cases. Each method was asked to place 20 decaps ($K = 20$) on each test.

### E.1  IMPEDANCE SUPPRESSION PLOTS

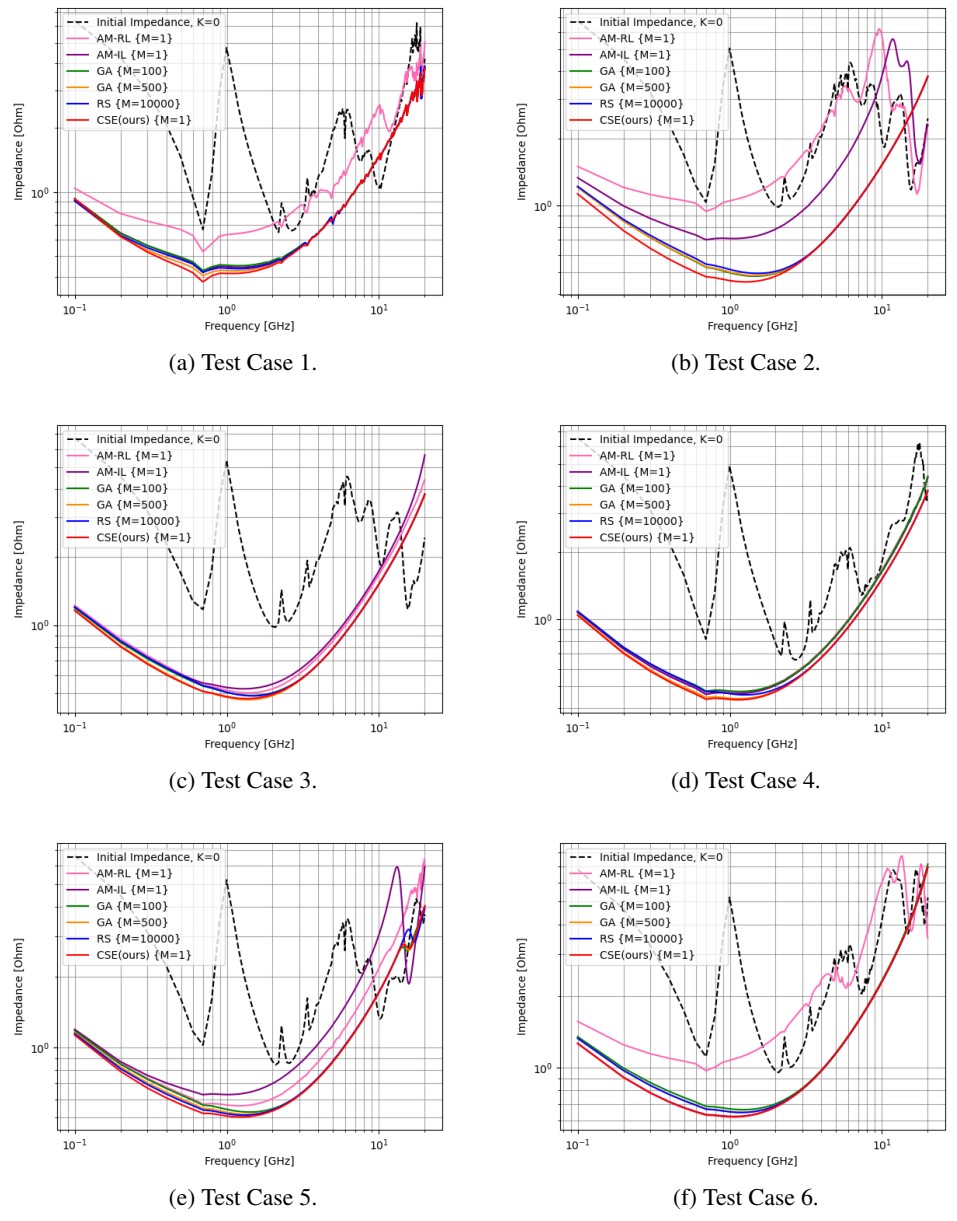

Figure 19: Impedance suppressed by each method, GA $\{M = 100\}$ (*expert policy*), GA $\{M = 500\}$, RS $\{M = 10,000\}$, AM-RL , AM-IL and CSE (Ours) for 6 example PDN cases out of 100 test dataset. (The lower the better.)

## E.2 DECAP PLACEMENT TENDENCY ANALYSIS

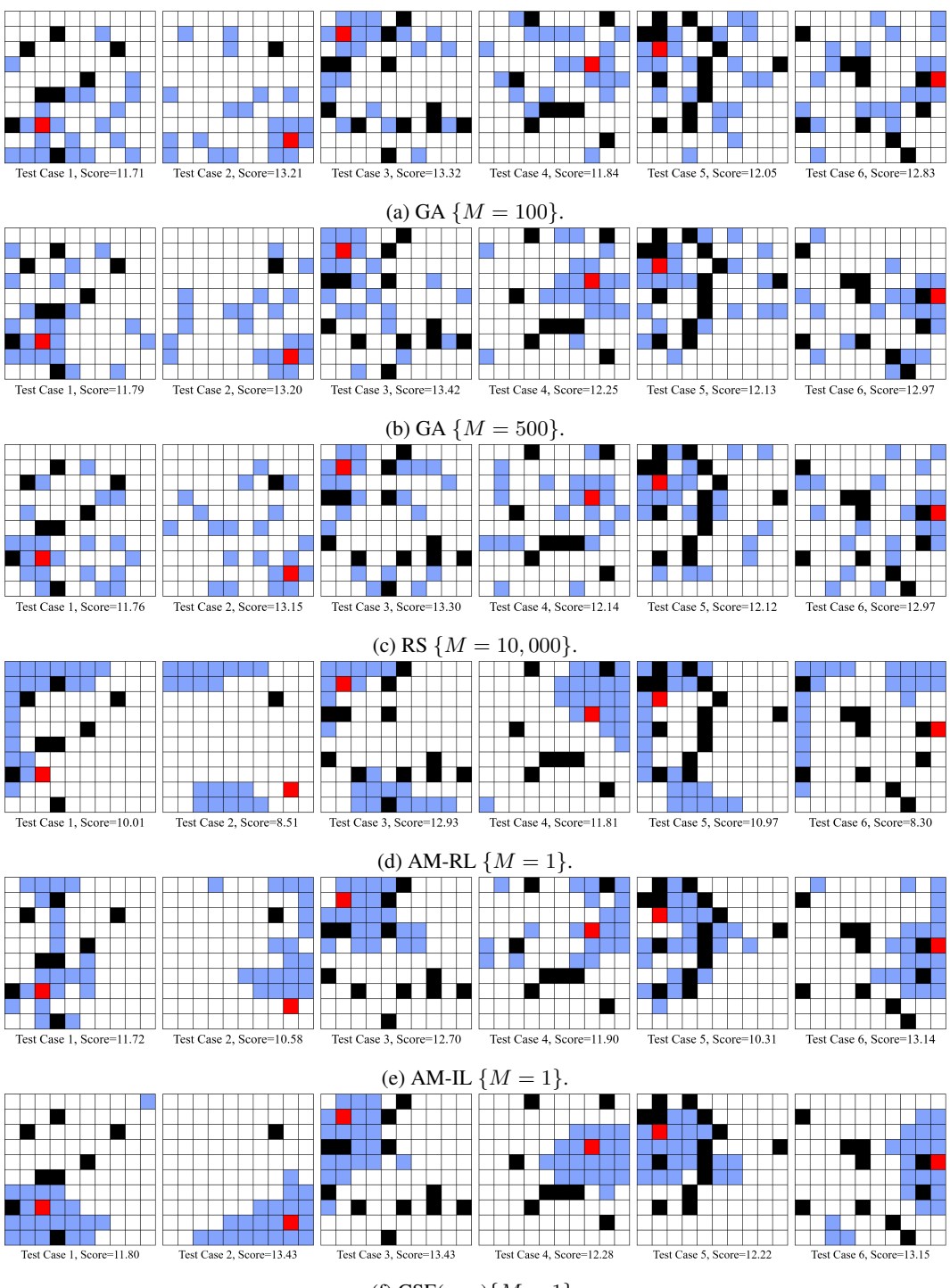

Figure 20: Corresponding decap placement solutions to Fig. 19 by each method. Red represents probing port, black represents keep-out ports and blue represents decap locations.

Fig. 21 shows the decap placement solutions of 6 PDN cases plotted in Fig. 19. The solutions by the search-heuristic methods, GA and RS, tend to be scattered while the solutions by learning-based methods, AM-RL, AM-IL and CSE, are clustered. Since search-heuristic methods are based on

random generations, they do not show clear tendency. On the other hand, learning based methods are based on a policy so that they have distinct tendency in placing decaps.

The role of placing decaps in hardware design is to decouple loop inductance of PDN. In terms of PI, analysis of loop inductance is critical, but at the same time, is complex (Farrahi & Koether, 2019). The loop inductance distribution of PDN highly depends on various design parameters such as the location of probing port, spacing between power/ground, size of PDN, and hierarchical layout of PDN (Fan et al., 2000). When human experts place decaps on PDN, there are too many domain rules to consider. On the other hand, CSE understands the PDN structure and its electrical properties by data-driven learning. According to Fig. 21, CSE tends to place decaps near the probing port, which is a well-known expert rule in the PI domain.

### E.3    POWER NOISE ANALYSIS ON HBM PDN

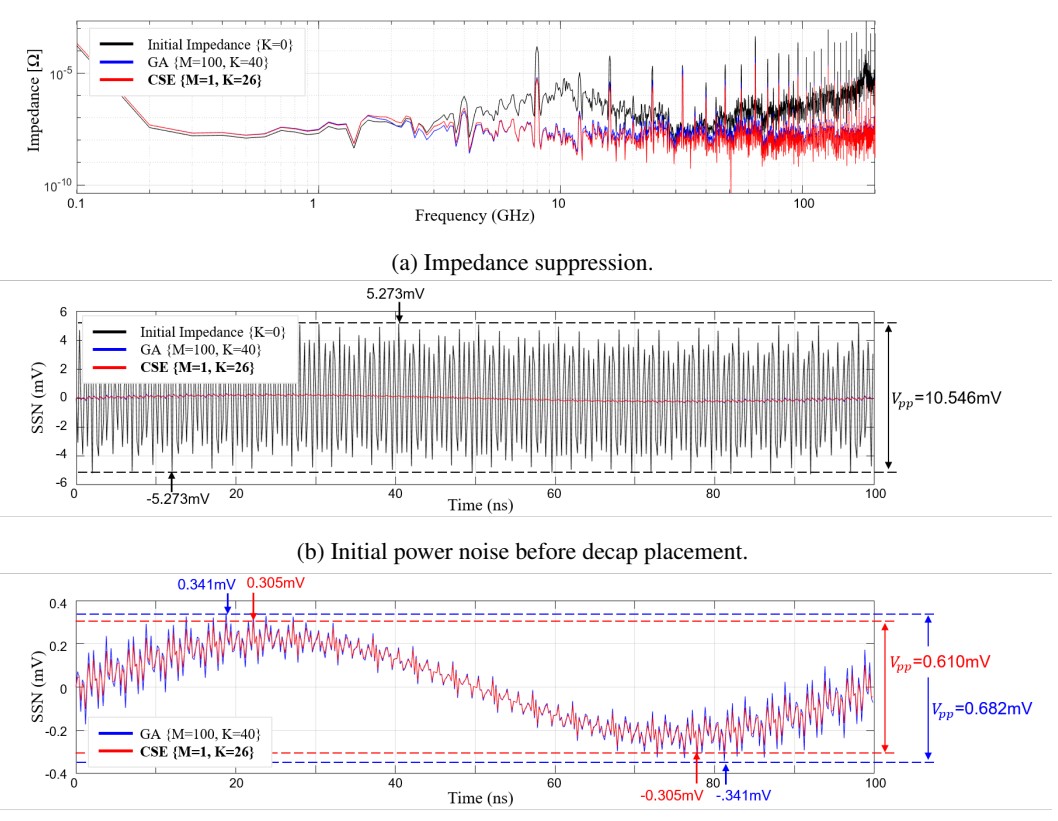

(a) Impedance suppression.

(b) Initial power noise before decap placement.

(c) Power noise after decap placement by our CSE$\{M = 1, K = 26\}$ and GA$\{M = 100, K = 40\}$.

Figure 21: Power noise analysis in terms of simultaneous switching noise (SSN) on HBM PDN before and after decap placed by our CSE$\{M = 1, K = 26\}$ and GA$\{M = 100, K = 40\}$.

Appendix E.3 analyzes the performance of CSE in comparison to GA$\{M = 100\}$ in terms of power noise. Out of 100 test cases on HBM PDN, we randomly chose a test case and carried out peak-to-peak power noise analysis for a circuit block, phase locked loop (PLL), operating at 5GHz. Note that CSE placed 26 decaps and GA$\{M = 100\}$ placed 40 decaps. CSE reduced power noise more than GA$\{M = 100\}$ with 14 less decaps. The impedances of the probing port on power distribution network (PDN) before and after decap placed by CSE and GA$\{M = 100\}$ are presented in Fig. 21a. The time-domain power noise before and after decap placement by each method is shown in Fig. 21b. For performance comparison, Fig. 21c shows the time-domain power noise after decap placement by CSE and GA$\{M = 100\}$.

# F  FURTHER ABLATION STUDY

This section reports ablation studies on action permutation invariance and hyperparameters $N$ (number of guiding samples), $\lambda$ (weight of self-exploitation loss term), and $P$ (number of permutation transformed labels).

## F.1  ABLATION STUDY ON $N$

$N$ is the number of expert labels generated by the expert policy, GA $\{M = 100\}$. We ablate $N \in \{100, 500, 1000, 2000\}$ with fixed $P = 3$ and $\lambda = 8$ and compare to AM-IL baseline for all $N$. As shown in Table 7, CSE with $N = 2000$ gives the best performance and CSE outperforms AM-IL for all $N$ variations. Performance of AM-IL is saturated at $N > 500$ while the performance of CSE continuously increases with the increase of $N$.

Table 7: Ablation study on $N$ for CSE ($P = 3, \lambda = 8$) and AM-IL.

|  | Validation Score |
| --- | --- |
| AM-IL $\{N = 100\}$ | 11.60 |
| CSE (ours) $\{N = 100\}$ | **12.98** |
| AM-IL $\{N = 500\}$ | 12.37 |
| CSE (ours) $\{N = 500\}$ | **12.99** |
| AM-IL $\{N = 1000\}$ | 12.23 |
| CSE (ours) $\{N = 1000\}$ | **13.09** |
| AM-IL $\{N = 2000\}$ | 12.32 |
| CSE (ours) $\{N = 2000\}$ | **13.13** |

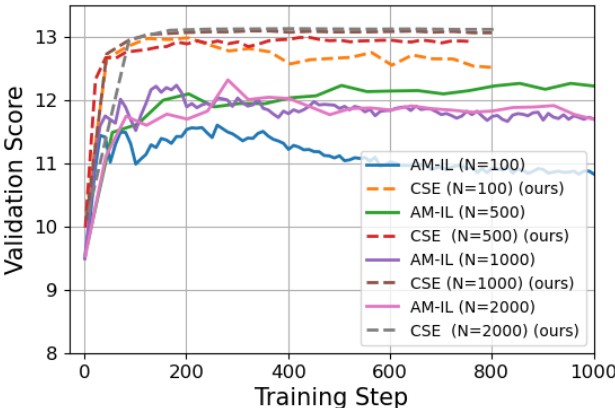

Figure 22: Validation graph CSE in comparison to AM-IL for varying number of offline expert data $N \in \{100, 500, 1000, 2000\}$.

## F.2 ABLATION STUDY ON $\lambda$

$\lambda$ refers to the weight of self-exploitation loss term $L_{Self}$, in the collaborative learning loss $\mathcal{L} := \mathcal{L}_{Expert} + \lambda \mathcal{L}_{Self}$. To set $\lambda \times L_U$ be $0.1 \sim 1$, we first multiplied $10^{32}$ to $\lambda$ because the probability of a specific solution is extremely small. Then, we ablated for $\lambda \in \{1, 2, 4, 6, 7, 8, 9, 10\}$ ($10^{32}$ is omitted) with fixed $N = 100$ and $P = 3$. For every $\lambda$, it prevents overfitting of the model in comparison to the baselines trained only with $L_{Expert}$ (see Fig. 23). According to the Table 9, $\lambda = 8$ gives the best validation scores.

Table 8: Ablation study of $\lambda$ on fixed $P = 3$ and $N = 100$.

| $\lambda$ ($\times 10^{32}$) | Validation Score |
|---|---|
| 1 | 12.96 |
| 2 | 12.96 |
| 4 | 12.94 |
| 6 | 12.96 |
| 7 | 12.98 |
| 8 | **12.98** |
| 9 | 12.97 |
| 10 | 12.96 |
| Only IL, $\lambda = 0$ | 12.97 |

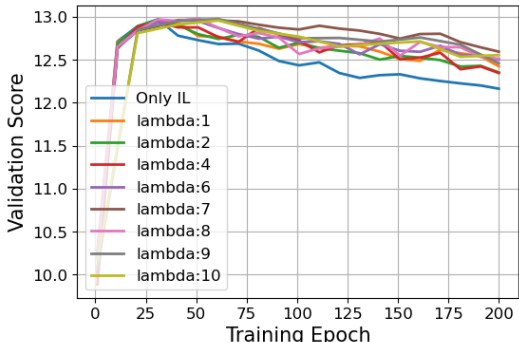

Figure 23: Validation graph of $\lambda \in \{1, 2, 4, 6, 7, 8, 9, 10\}$ on fixed $P = 3$ and $N = 100$.

### F.3    ABLATION STUDY ON $P$

$P$ is the number of permutation transformed labels per each expert label used for imitation learning-based expert exploitation. We ablate $P \in \{3, 5, 7\}$ with fixed $N = 100$ and $\lambda = 8$ and compared collaborative symmetricity exploitation (i.e., both expert and self-exploitation) to only expert exploitation training case. As shown in Table 9, $P = 3$ with {Expert exploitation + Self-exploitation} give best performances. For every $P$, {Expert exploitation + Self-exploitation} gives the better performances, indicating self-exploitation scheme well prevents overfitting of training process for sparse dataset.

Table 9: Ablation study on $P$ with and without unsupervised loss term.

|  | Validation Score |
|---|---|
| Expert exploitation $\{P = 3\}$ | 12.97 |
| + Self- exploitation $\{\lambda = 8\}$ | **12.98** |
| Expert exploitation $\{P = 5\}$ | 12.95 |
| + Self- exploitation $\{\lambda = 8\}$ | **12.95** |
| Expert exploitation $\{P = 7\}$ | 12.93 |
| + Self- exploitation $\{\lambda = 8\}$ | **12.95** |

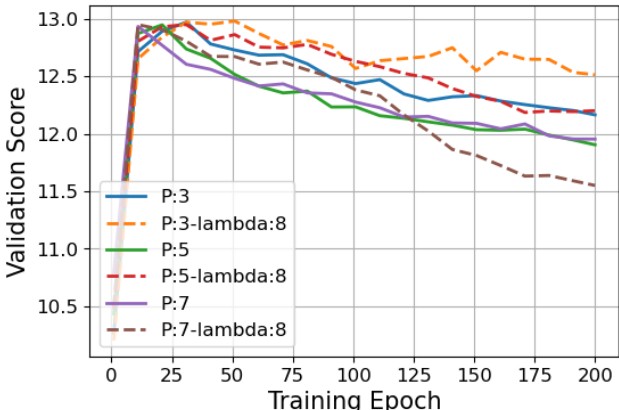

Figure 24: Validation score of $P$ ablation with and without self-exploitation loss term.

## G    PROOF OF THEOREM 3.1

($\rightarrow$) Suppose that policy $\pi(\boldsymbol{a}|\boldsymbol{x})$ is AP-symmetric. Then, by the Definition 3.1, $\pi(\boldsymbol{a}|\boldsymbol{x}) = \pi(t(\boldsymbol{a})|\boldsymbol{x})$ for any $\boldsymbol{a} \in \mathcal{A}$, $\boldsymbol{x} \in \mathcal{X}$, $t \in T_{AP}$.

Therefore,

$$b(\pi; \boldsymbol{p}) = \mathbb{E}_{\boldsymbol{x} \sim p_{\mathcal{X}}(x)} \mathbb{E}_{\boldsymbol{a} \sim p_{\mathcal{A}}(\boldsymbol{a})} \mathbb{E}_{t \sim p_{T_{AP}}(t)}[||\pi(\boldsymbol{a}|\boldsymbol{x}) - \pi(t(\boldsymbol{a})|\boldsymbol{x})||_1] = 0$$

($\leftarrow$) Suppose that $b(\pi; \boldsymbol{p}) = 0$, where $p_{\mathcal{X}}(\boldsymbol{x}) > 0, p_{\mathcal{A}}(\boldsymbol{a}) > 0, p_{T_{AP}}(t) > 0$.

Assume that there exist $\boldsymbol{a}^* \in \mathcal{A}$, $\boldsymbol{x}^* \in \mathcal{X}$, and $t^* \in T_{AP}$, such that $\pi(\boldsymbol{a}^*|\boldsymbol{x}^*) \neq \pi(t(\boldsymbol{a}^*)|\boldsymbol{x}^*)$.

Then,

$$b(\pi; \boldsymbol{p}) = \mathbb{E}_{\boldsymbol{x} \sim p_{\mathcal{X}}(x)} \mathbb{E}_{\boldsymbol{a} \sim p_{\mathcal{A}}(\boldsymbol{a})} \mathbb{E}_{t \sim p_{T_{AP}}(t)}[||\pi(\boldsymbol{a}|\boldsymbol{x}) - \pi(t(\boldsymbol{a})|\boldsymbol{x})||_1]$$
$$\geq p_{\mathcal{X}}(\boldsymbol{x}^*) p_{\mathcal{A}}(\boldsymbol{a}^*) p_{T_{AP}}(t^*)||\pi(\boldsymbol{a}^*|\boldsymbol{x}^*) - \pi(t(\boldsymbol{a}^*)|\boldsymbol{x}^*)||_1 > 0,$$

which results in a contradiction. Therefore, $\pi(\boldsymbol{a}|\boldsymbol{x}) = \pi(t(\boldsymbol{a})|\boldsymbol{x})$ for any F$\boldsymbol{a} \in \mathcal{A}$, $\boldsymbol{x} \in \mathcal{X}$, $t \in T_{AP}$: i.e, policy $\pi(\boldsymbol{a}|\boldsymbol{x})$ is AP-symmetric.

## H    ORDER BIAS MEASUREMENT

This section reports the order bias measurements of AM-IL and our CSE. We measured $b(\pi, \boldsymbol{p} = \{\mathcal{U}_{\mathcal{X}}, \pi, \mathcal{U}_{T_{AP}}\}$ for sample width 100 and took the average value. As shown in Appendix H, our CSE significantly reduced the order bias, verifying that CSE successfully induced the AP-symmetricity.

Table 10: Evaluation of Order Bias

|  | $b(\pi, \boldsymbol{p} = \{\mathcal{U}_{\mathcal{X}}, \pi, \mathcal{U}_{T_{AP}}\})$ |
|---|---|
| AM-IL | $8.70 \times 10^{-21}$ |
| **CSE (ours)** | $\mathbf{1.25} \times 10^{-28}$ |

