# OpenReview forum: "Collaborative Symmetricity Exploitation for Offline Learning of Hardware Design Solver"
_ICLR.cc/2023/Conference — Submitted to ICLR 2023_

### Official Review · Reviewer_1Sai · 2022-10-22

**Confidence:** 3
**Correctness:** 3
**Technical Novelty And Significance:** 3
**Empirical Novelty And Significance:** 3
**Recommendation:** 6

**Clarity, Quality, Novelty And Reproducibility:**

The paper is not clear enough for those who are not very familiar with imitation learning. Notations in Figure 2&4 are not easy to follow. The idea to use symmetry seems novel in DDP.

**Strength And Weaknesses:**

Strength:
1. A new imitation learning framework for hardware tasks, featuring zero-shot, low requirements for training data.
2. Experiment results seem good.
3. Sufficient supplementary materials.

Weakness:
1. Figure 2&4 are not clear enough. It is recommended to explain the complete process of the entire framework in the main text.
2. Further explanations should be given on the reason why the CSE policy trained with low-quality offline expert data can produce higher-quality designs if possible.
3. Typo in Figure 1.


**Summary Of The Paper:**

The paper proposes a collaborative symmetricity exploitation framework to solve the decoupling capacitor placement problem. The framework leverages action-permutation symmetry in offline imitation learning to augment data and to improve generalization capability. It can outperform iterative online search methods like GA in DDP with zero-shot inference.

**Summary Of The Review:**

The paper provides a novel framework for DDP. The framework is impressive considering its performance and data-efficiency. Some parts can be improved.

---

> ### Author Response · Authors · 2022-11-16
> **Responses**
>
> Thank you for your valuable comments. We revised our manuscript according to your comments.
>
> ---
>
> **Response to the Weaknesses**
>
> * We revised Figure 2 and 4 in the main paper.
>
> * We also revised section 5.2 of the main paper based on your comment. CSE was capable of extrapolation over low-quality offline data due to the following reasons. First, an general context about AP-symmetricity was explicitly given to the model. Second, contextualized neural architecture was capable to extract major context to determine DPP performances. Lastly, policy generates a solution with high confidence (with greedy rollout) based on extracted context.
>
> * Typo in Figure 1 is corrected.
>
> ---
>
> **Clarity of the learning scheme**
>
> To improve the clarity of the learning scheme we newly introduce the order-bias metric and theorem which mathematically explains loss function design.

---

### Official Review · Reviewer_PJWA · 2022-10-23

**Confidence:** 3
**Correctness:** 4
**Technical Novelty And Significance:** 3
**Empirical Novelty And Significance:** 2
**Recommendation:** 5

**Clarity, Quality, Novelty And Reproducibility:**

The writing of this paper is clear and the quality is good. The novelty of this paper is relatively good and the experiments are reproducible.

**Details Of Ethics Concerns:**

There are no ethics concerns.

**Strength And Weaknesses:**

Strength:
1. The major challenge for solving such placement problems is the combinatorial explosion.  The idea of applying solution symmetricity is interesting and can potentially reduce the problem difficulty significantly.
2. The writing of this paper is good and the presentation is clear.
3. The considered problem is theoretically formulated as an MDP so that the similar idea might be easily applicable to other scenarios.

Weaknesses:
1. In the decap placement problem considered here, the order of placement of each decap is not important, which is assumed in this paper. Whether the CSE idea is still applicable somehow if the order of placement of each element matters?
2. The evaluation of the efficiency of the proposed method is somewhat quite sparse, only comparison with the AM-IL baseline. With these inadequate evaluations, it's not very convincing that how the proposed approach is more efficient than other baselines.
3. The novelty of this paper mainly lies in the symmetricity exploitation. The design choices of the recurrent neural architecture are quite similar to typical architectures for tackling such placement problems.

**Summary Of The Paper:**

This paper has proposed the idea of collaborative symmetricity exploitation (CSE) that merges all AP-symmetric solution trajectories of placement problems into a single merged trajectory. This idea can reduce the search space and is applied to tackle the decoupling capacitance placement problem. Specifically, this paper has devised a neural network model to generate the sequential placement action based on the existing placement state. The proposed approach is empirically compared against reinforcement learning RL or imitation learning (IL) baselines. The proposed approach shows better efficiency than baselines.

**Summary Of The Review:**

The idea of utilizing the solution symmetricity for certain placement problems is interesting. The proposed method is sound and has potential to be applied in industrial scenarios. However, more empirical evaluations are required to fully demonstrate the advantages of the proposed method over other baselines. Also, it's good to show some perturbation analysis of hyper-parameters.

---

> ### Author Response · Authors · 2022-11-16
> **Reponses**
>
> Thank you for your valuable comments. We'd like to address your comments by providing further explanations and additional experimental results.
>
> ---
>
> **Response to the Weaknesses**
>
> * Our CSE is applicable to any placement problem, where the order of placement does not affect the overall performance. We define placement design task as assigning components to generate final design. For instance, when we were given furniture placement task, the final design is what matters rather than in what order the furniture were placed. If the order of placement matters, the task is rather considered a scheduling or routing problem such as TSP.
>
> * We conducted additional experiments on sample-efficiency of Arb-IL and reported in revised Figure 6. Since the sample refers to offline data, we only compared with IL-based baselines. As a result, CSE was proven most sample-efficient (i.e., showed highest performance with much less offline data) out of IL methods. Furthermore, the performance of AM-IL and Arb-IL saturated at $N=500$ while the performance of CSE continuously increased with $N$.
>
> ---
>
> **Technical Novelty**
>
> Learning symmetricity in combinatorial space is important and difficult. To achieve AP-symmetricity (which is the general symmetric nature of placement tasks), we proposed a simple and novel loss scheme, which gives both empirical powerful performances and theoretical analysis.
>
> To clarifies the technical novelty of our loss scheme, we formally introduced the order-bias metric to measure AP-symmetricity in the revised manuscript. The order bias is a general property of a sequential solution generation scheme. It measures how much the solver has different probabilities to generate AP-symmetric solutions. We theoretically proved that AP-symmetricity is satisfied when order bias becomes zero (**Theorem 3.1**).
>
> Our loss scheme is specifically designed to reduce the order bias based upon theoretical proof that the designed loss term guarantees AP-symmetricity. Therefore, our technical novelty comes from designing a proper loss scheme using the theoretical analysis, not from applying self-supervised learning.
>
> ---
>
> **Empirical Evaluation and Perturbation Analysis**
>
> Empirical evaluation of the proposed method in terms of power noise reduction on high bandwidth memory in comparison to the baseline method is reported in **Appendix E**. Sensitivity analysis on hyperparameters is provided in **Appendix F**.

---

### Official Review · Reviewer_XtSs · 2022-10-25

**Confidence:** 3
**Correctness:** 3
**Technical Novelty And Significance:** 2
**Empirical Novelty And Significance:** 2
**Recommendation:** 3

**Clarity, Quality, Novelty And Reproducibility:**

The paper is clearly written and of decent quality.

The paper lacks novelty and reproducibility. The paper does not open source the code or benchmark used.

**Strength And Weaknesses:**

Strengths:
- The leverage of symmetricity in placement problems can effectively improve data-efficiency of training and generalization over changing task conditions.


Weaknesses:
- The task of decoupling capacitance placement problem is not something the general community are familiar with. The proposed method can be very useful for that particular problem, but the generalization is questionable unless it can be evaluated on other tasks or domains.

- The proposed method is not novel as policy gradient based reinforce and self-supervised learning are not new.

**Summary Of The Paper:**

This paper proposes a collaborative symmetricity exploitation framework to train a solver for the decoupling capacity placement problem benchmark. Since hardware design is multi-level and sequential and performance simulation is costly, data-efficient offline learning with good generalization becomes critical. This paper applies CSE to train a DPP solver using a limited number of offline expert data. Expoert exploitation induces symmetricity during the limited learning process and self-exploitation induces symmetricicty during the consistency learning process with self-generated data.

**Summary Of The Review:**

The application in this paper is something not many people are aware of, which may penalize the paper. But the reviewer still feels that a machine learning paper should aim for general benchmarks that many of us are more familiar with. The reviewer understands that there are prior works getting traction by applying old techniques to a new domain (computer systems, circuit design, etc.), but those works did a great job promoting a new field of applications so that more and more researchers can build work on top of theirs.

in order to improve this work, consider:
- Promoting the application in this paper and make sure more people in this community are aware of it and would love to work on methods to improve the state-of-the-arts.

- Be clear which parts are novel and which parts are compilation of existing work. If the proposed method is novel, be very specific in the introduction and abstract.

- Create more thorough evaluation with more benchmarks and stronger baselines. Explain in details why the baselines are not good enough.

---

> ### Author Response · Authors · 2022-11-16
> **Regarding DPP Benchmark and reproducibility**
>
> Thank you for your valuable comments. We'd like to address your comments by providing further explanations. Moreover, we revised our manuscript based on your comment and concern. We sincerely request you read our improved manuscript as your comment was extremely helpful to improve our paper.
>
> ---
>
> **Regarding DPP Benchmark**
>
> We admit that DPP is not something the ML community is familiar with and we know there exist well-explored benchmarks (ex., design-bench [1]) that are worth solving. However, DPP is a new type of task and has distinct characteristics that widely-solved benchmarks do not have any characteristics that are shared with various real-world problems, especially hardware problems.
>
> Characteristics of DPP benchmark are:
>
> * Objective evaluation and data generation are costly in terms of time and computation; less iteration is preferred and high sample-efficiency is required (this is a common thing shared with other offline black-box optimization benchmarks [1] ).
>
> * Task condition is determined by higher-level task and is changing; a task-conditioned contextualized policy (i.e., solver) capable of producing solutions to changing problem conditions by zero-shot inference is necessary (**this is a novel characteristic of DPP benchmark**).
>
> To this end, we sincerely request to rethink our DPP benchmark which is not only a specific micro-electrical design benchmark but also a general ML benchmark of **Offline Contextual Black-box Optimization**.
>
> The novelty of our method CSE lies mainly in designing symmetric learning for contextual policy in an offline manner. In fact, many tasks in the hardware domain, such as chip placement, via placement and routing, share the characteristics mentioned above and thus require ML methodologies. CSE is applicable to any placement task and will give promising performances as a general method of offline contextual black-box optimization.
>
> ---
>
> **Reproducibility**
>
> We will open our source codes for CSE and the DPP benchmark public after decision is made. We will encourage researchers in the ML community to further engage in solving DPP benchmark to develop various algorithms applicable to real-world problems.
>
> ---
>
> [1] Trabucco, Brandon, et al. "Design-bench: Benchmarks for data-driven offline model-based optimization." in ICML 2022.

---

> > ### Author Response · Authors · 2022-11-16
> > **Technical Novelty and State-of-the-art Performances**
> >
> > **Clarification of Technical Novelty**
> >
> > To clarifies the technical novelty of our loss scheme, we formally introduced the order-bias metric to measure AP-symmetricity in the revised manuscript. The order bias is a general property of a sequential solution generation scheme. It measures how much the solver has different probabilities to generate AP-symmetric solutions. We theoretically proved that AP-symmetricity is satisfied when order bias becomes zero (**Theorem 3.1**).
> >
> > Our loss scheme is specifically designed to reduce the order bias based on theoretical proof that the designed loss term guarantees AP-symmetricity. Therefore, our technical novelty comes from designing a proper loss scheme using theoretical analysis, not from simply applying self-supervised learning.
> >
> > ---
> >
> > **State-of-the-art Performance and Baselines**
> >
> > There were several machine learning-based works to improve the optimality of DPP. It became a highly competitive task recently in the hardware design community (without any reproducible codes and resources). Part et al. [1]  recently benchmarked seven DRL-based baseline methods [2,3,4,5,6,7,8], and claimed state-of-the-art and was verified to be published in the top-tier hardware journal (IEEE Transactions on MTT; 2022).
> >
> > In this work, we focus to reproduce the most competitive baselines (AM-RL [1], Arb-RL [2]), which can give sample efficiency in the inference phase with contextual policy training. Note that reproducing hardware design baseline is extremely challenging (to compare with typical ML baselines) because (a) they usually do not open the source code (b) the algorithmic description is highly focused on the hardware application and electrical model. According to the experimental results, our CSE outperforms the state-of-the-art DRL model [1].
> >
> > Moreover, we also provided a search-based meta-heuristic as the baseline and evaluate our method in an extremely harsh environment: CSE (zero-shot) vs. GA (500 simulations). As shown in Table 1, our CSE takes 500 fewer simulations per given problem and still outperforms GA.
> >
> >
> >
> > ---
> >
> > [1] H. Park et al., "Transformer Network-Based Reinforcement Learning Method for Power Distribution Network (PDN) Optimization of High Bandwidth Memory (HBM)," in IEEE Transactions on Microwave Theory and Techniques, vol. 70, no. 11, pp. 4772-4786, Nov. 2022, doi: 10.1109/TMTT.2022.3202221.
> >
> > [2] H. Kim et al., “Deep reinforcement learning framework for optimal decoupling capacitor placement on general PDN with an arbitrary probing port,” in Proc. IEEE 30th Conf. Electr. Perform. Electron. Packag. Syst. (EPEPS), Oct. 2021, pp. 1–3.
> >
> > [3] J. Shin et al., “Reinforcement learning-based decap optimization method for high-performance solid-state drive,” in Proc. IEEE Int. Joint EMC/SI/PI EMC Eur. Symp., Jul. 2021, pp. 718–721.
> >
> > [4] L. Zhang et al., “Decoupling capacitor selection algorithm for PDN based on deep reinforcement learning,” in Proc. IEEE Int. Symp. Electromagn. Compat., Signal Power Integrity, Jul. 2019, pp. 616–620.
> >
> > [5] H. Park et al., “Deep reinforcement learning-based optimal decoupling capacitor design method for silicon interposer-based 2.5-D/3-D ICs,” IEEE Trans. Compon., Packag., Manuf. Technol., vol. 10, no. 3, pp. 467–478, Mar. 2020.
> >
> > [6] L. Zhang, W. Huang, J. Juang, H. Lin, B.-C. Tseng, and C. Hwang, “An enhanced deep reinforcement learning algorithm for decoupling capacitor selection in power distribution network design,” in Proc. IEEE Int. Symp. Electromagn. Compat. Signal/Power Integrity (EMCSI), Jul. 2020, pp. 245–250.
> >
> > [7] S. Han, O. W. Bhatti, and M. Swaminathan, “Reinforcement learning for the optimization of decoupling capacitors in power delivery networks,” in Proc. IEEE Int. Joint EMC/SI/PI EMC Eur. Symp., Jul. 2021, pp. 544–548.
> >
> > [8]  H. Park et al., “Policy gradient reinforcement learning-based optimal decoupling capacitor design method for 2.5-D/3-D ICs using transformer network,” in Proc. IEEE Electr. Design Adv. Packag. Syst. (EDAPS), Dec. 2020, pp. 1–3.

---

### Official Review · Reviewer_1PDU · 2022-10-26

**Confidence:** 2
**Clarity, Quality, Novelty And Reproducibility:** This work is of good quality.
**Correctness:** 3
**Technical Novelty And Significance:** 2
**Empirical Novelty And Significance:** 2
**Recommendation:** 5

**Strength And Weaknesses:**

Strength:
- This paper is well organised and easy to follow. The proposed design is thoroughly introduced.
- Related works are well covered and discussed.
- Experimental results have backed the claims and suggests good performance in real-world applications

Weaknesses:
- Compare to "Collaborative Distillation Meta Learning for Simulation Intensive Hardware Design" https://arxiv.org/abs/2205.13225, which also based on some collaborative approach and attention model, what extra novelty does this work has brought?
- Experimental results could be more convincing, Is there any benchmark on the practical application (5.5) comparing against other methods like AM or ArB? On top of the previous comment, why does the proposed work have the same stability test result as the referenced work?
- A minor note,  figure one has the same branches placement subfigure. It could be my misunderstanding, but if so this can still benefit from more explanation.



**Summary Of The Paper:**

This paper proposes a framework for decoupling capacitor placement problem including.

**Summary Of The Review:**

An interesting idea for a specific application, but I am not sure if it brings enough novelty and broader applicability.

---

> ### Author Response · Authors · 2022-11-16
> **Responses to the Weaknesses**
>
> Thank you for your valuable comments. We'd like to address your comments by providing further explanations and additional experimental results.
>
> ---
>
> **Responses to the Weaknesses**
>
> * The paper you referenced is a work-in-progress paper, which has not yet been published. Please consider there is some possibility that the work-in-progress paper is finally submitted to this ICLR 2023 venue as our CSE paper. Please understand we cannot provide any further information due to the double-blind policy of ICLR.
>
> * Based on your feedback, we provide extra experimental results for AM-IL and Arb-IL. IL-based methods including CSE were trained with $N=1,000$ offline data. See our revised Figure 8 and section 5.5. We compared the CSE to GA\{$M=100$\}, AM-IL and Arb-IL on placing varying number of decaps, $K$, for 100 test cases. CSE achieved the highest performance with significantly fewer decaps. For instance, the performance score of 26.68, attained with 40 decaps by GA\{$M=100$\} and 34 decaps by AM-IL, was achieved with only 26 decaps placed by zero-shot CSE. Arb-IL was unable to achieve 26.68 even with 80 decaps. Reducing the number of decaps is a huge contribution to the industry as hardware devices are mass-produced, reducing a single decap can greatly reduce production costs and also saves space for other components.
>
> * The placement subfigures on figure 1 are intended to be the same. We wanted to emphasise that there exists many trajectories that result in identical placement solution as the order of placement does not matter for hardware placement tasks. Figure 1 elucidates our motivation for designing a policy that is not affected by the order and, at the same time, for reducing solution space. To improve the clarification of figure 1, we revised the figure at the new manuscript.

---

> > ### Author Response · Authors · 2022-11-16
> > **Novelty and Further Applications**
> >
> > Inducing symmetricities to improve generalization capability of model has widely been practiced in various fields such as point cloud regression and molecule graph [1,2,3,4]. However, such methods induce symmetricity to the input of neural network because they target regression and classification tasks, which do not generate complex solutions. On the other hand, inducing symmetricities to the output solution space of neural network is less explored and is very challenging.
> >
> > Placement tasks such as our DPP find solutions from combinatorial space. Thus, auto-regressive schemes were widely implemented due to its capability of processing a high-dimensional input. However, auto-regressive schemes incur unwanted inductive bias because sequentially generating solution leads to unwanted bias in terms of the order of placement.
> >
> > GflowNet [5] was proposed to overcome the limitation in molecular generation task by inducing solution symmetricities. However, GflowNet optimizes a single task while DPP requires a task-conditioned contextualized policy (i.e., optimizing infinite tasks with different conditions). Our CSE allows training of a task-conditioned contextualized policy while inducing solution symmetricities to improve sample efficiency of training and generalization capability of the trained policy.
> >
> > Though we only applied to DPP, the proposed CSE can be applied to various applications including chip placement, via placement, floor planning, robot assignment and set covering as long as sufficient expert offline data is available.
> >
> > ---
> >
> > [1] Taco Cohen and Max Welling. Group equivariant convolutional networks. In International conference on machine learning, pp. 2990–2999. PMLR, 2016.
> >
> > [2] Nathaniel Thomas, Tess Smidt, Steven Kearnes, Lusann Yang, Li Li, Kai Kohlhoff, and Patrick Riley. Tensor field networks: Rotation-and translation-equivariant neural networks for 3d point clouds. arXiv preprint arXiv:1802.08219, 2018.
> >
> > [3] Fabian Fuchs, Daniel Worrall, Volker Fischer, and Max Welling. Se (3)-transformers: 3d rototranslation equivariant attention networks. Advances in Neural Information Processing Systems, 33:1970–1981, 2020.
> >
> > [4] Victor Garcia Satorras, Emiel Hoogeboom, and Max Welling. E(n) equivariant graph neural networks. In International conference on machine learning, pp. 9323–9332. PMLR, 2021.
> >
> > [5] Emmanuel Bengio, Moksh Jain, Maksym Korablyov, Doina Precup, and Yoshua Bengio. Flow network based generative models for non-iterative diverse candidate generation. Advances in Neural Information Processing Systems, 34:27381–27394, 2021.

---

### Author Response · Authors · 2022-11-16
**General Comment to the Reviewers**



We would first like to appreciate all your valuable comments. We revised our paper based on your comments and marked them in blue.

In the revised paper, we formally defined the order-bias metric with a theoretical analysis of AP-symmetricity (**Appendix G**). To this end, we provide some reasoning for the loss function design of CSE using a formally defined order-bias metric. Moreover, we report empirical measurement of AP-symmetricity based on the order-bias metric in **Appendix H**. We expect this revision process can improve the clarity of the method and technical novelty. We also revised the abstract, figure 1, and the introduction, experimental results section based on the reviewers' comments.

The source codes for the DPP benchmark and CSE as well as the baselines will be released to the public after the decision is made. Here we present clarifications to the novelty and contribution of this work.

---
**A novel symmetric learning scheme for contextualized policy**

There exists several works [1,2,3,4] that learn various symmetricities of input data in the domain space for regression and classification tasks. However, learning symmetricity in solution space is less studied as learning the symmertrcities in solution space of sequential policy (generative decision) is challenging. [5] tackled solution symmetricity of sequential policy by turning the Markov decision process (MDP) tree model into the directed acyclic graph (DAG)-based flow model. However, they target single-task optimization where the optimal solution set is unchanged. On the other hand, our CSE is an effective solution symmetric learning scheme for the contextualized policy capable of adapting to newly given task-condition.

---

**DPP benchmark**

DPP is a widely studied task in hardware domain without public release of the simulation models and source codes for the methods. Also, DPP can be seen as a contextual offline black-box optimization benchmark with extended properties compared to the design-bench [6], a representative non-contextual offline black-box optimization benchmark. In this work, by releasing the DPP benchmark with open-source simulation models and our reproduced baselines, DRL-based methods, meta-heuristic methods, behavior cloning-based methods, and our state-of-the-art CSE method, we expect huge industrial impacts on the hardware and the ML communities.

---

[1] Taco Cohen and Max Welling. Group equivariant convolutional networks. In International conference
on machine learning, pp. 2990–2999. PMLR, 2016.

[2] Nathaniel Thomas, Tess Smidt, Steven Kearnes, Lusann Yang, Li Li, Kai Kohlhoff, and Patrick Riley.
Tensor field networks: Rotation-and translation-equivariant neural networks for 3d point clouds.
arXiv preprint arXiv:1802.08219, 2018.

[3] Fabian Fuchs, Daniel Worrall, Volker Fischer, and Max Welling. Se (3)-transformers: 3d rototranslation equivariant attention networks. Advances in Neural Information Processing Systems,
33:1970–1981, 2020.

[4] Victor Garcia Satorras, Emiel Hoogeboom, and Max Welling. E(n) equivariant graph neural networks.
In International conference on machine learning, pp. 9323–9332. PMLR, 2021.

[5] Emmanuel Bengio, Moksh Jain, Maksym Korablyov, Doina Precup, and Yoshua Bengio. Flow
network based generative models for non-iterative diverse candidate generation. Advances in
Neural Information Processing Systems, 34:27381–27394, 2021.

[6] Trabucco, Brandon, et al. "Design-bench: Benchmarks for data-driven offline model-based optimization." in ICML 2022.

---

### Author Response · Authors · 2022-12-05
**Urgent request for discussion**

**Dear AC and reviewer,**

Today is about a week before the end of the discussion. Thus, we kindly remind reviewers and AC to participate in the discussion for our responses.

We tried to resolve concerns in both theoretical and empirical ways. Also, there are still many points to discuss or debate to qualify our paper. Furthermore, this forum will be open to every researcher; the discussion process between author and reviewer may be helpful to communities regardless of its' acceptancy. We hope reviewers and AC participate actively in the remaining discussion period.

**Sincerely,**

**Authors**

---

> ### Comment · Area_Chair_REBx · 2022-12-05
> **the status of discussion**
>
> Dear authors,
>
> There has already been a discussion between the AC and the reviewers, which can not be seen by the authors. If you request a discussion or feedback for a specific reviewer, you can directly message him/her.
>
> Thanks,
>
> AC

---

### Decision · Program_Chairs · 2023-01-20

**Decision:**

Reject

**Justification For Why Not Higher Score:**

The major concern is that the proposed technique needs to show its effectiveness in other domains.

**Justification For Why Not Lower Score:**

N/A

**Metareview: Summary, Strengths And Weaknesses:**

In this paper the authors propose a so-called collaborative symmetricity exploitation (CSE)  scheme to train contextualized policies and apply it to the decoupling capacity placement problem (DPP).  The authors argue that exploiting symmetricity can improve data efficiency and help generalization. The authors also show the state-of-the-art result on the DPP benchmark to demonstrate the effectiveness of the proposed CSE approach.  In the rebuttal, the authors addressed most of the concerns raised by the reviewers and significantly revised the original submission.  While the rebuttal and revision have greatly improved the quality of the paper and clarified some of the concerns, a number of issues still stand.  First of all, the paper may have a good impact to the DPP domain with its decent performance and a good theoretical framework. However, as a technique that is claimed to be task-agnostic, it would make a much stronger case if the authors can show its performance on tasks other than the hardware design and hence show it indeed improves generalization capability in other domains.   In addition, the modifications made in the revision seem to be a bit significant compared to the original submission  (e.g. reclaimed contributions, one added theorem etc.).  Based on the discussion among the reviewers, the paper can not be accepted in its current form.  I would suggest the authors strengthen the experiments and submit it to a future conference.